# Microglia control vascular architecture via a TGFβ1 dependent paracrine mechanism linked to tissue mechanics

Tejasvi Dudiki[1], Julia Meller[1], Gautam Mahajan[2], Huan Liu[1], Irina Zhevlakova[1], Samantha Stefl[1,4], Conner Witherow[1,4], Eugene Podrez[3], Chandrasekhar R. Kothapalli[2] & Tatiana V. Byzova [1✉]

Tissue microarchitecture and mechanics are important in development and pathologies of the Central Nervous System (CNS); however, their coordinating mechanisms are unclear. Here, we report that during colonization of the retina, microglia contacts the deep layer of high stiffness, which coincides with microglial bipolarization, reduction in TGFβ1 signaling and termination of vascular growth. Likewise, stiff substrates induce microglial bipolarization and diminish TGFβ1 expression in hydrogels. Both microglial bipolarization in vivo and the responses to stiff substrates in vitro require intracellular adaptor Kindlin3 but not microglial integrins. Lack of Kindlin3 causes high microglial contractility, dysregulation of ERK signaling, excessive TGFβ1 expression and abnormally-patterned vasculature with severe malformations in the area of photoreceptors. Both excessive TGFβ1 signaling and vascular defects caused by Kindlin3-deficient microglia are rescued by either microglial depletion or microglial knockout of TGFβ1 in vivo. This mechanism underlies an interplay between microglia, vascular patterning and tissue mechanics within the CNS.

───────────────
[1] Department of Neurosciences, Lerner Research Institute, Cleveland Clinic, Cleveland, OH, USA. [2] Chemical and Biomedical Engineering Department, Washkewicz College of Engineering, Cleveland State University, Cleveland, OH, USA. [3] Department of Inflammation and Immunity, Lerner Research Institute, Cleveland Clinic, Cleveland, OH, USA. [4] Present address: Department of Cardiovascular and Metabolic Sciences, Lerner Research Institute, Cleveland Clinic, Cleveland, OH, USA. ✉email: byzovat@ccf.org

Tissue architecture, which is determined by complex and dynamic interactions between various cell types and the extracellular matrix, and tissue mechanics are reciprocally connected in health and pathologies[1]. While the role of these interactions has been demonstrated in cell differentiation and malignancies, very little is known about it in the central nervous system (CNS) because of its complex cellular composition and mechanical heterogeneity[2,3]. Thinning of retinal layers in Alzheimer's patients[4] alters retinal stiffness and correlates with the changes in microvasculature that mirror pathological processes in the brain[5]; CNS scarring and neurodegeneration are accompanied by similar changes[6–8]. Compared with other tissues, the retina seems to be particularly affected by a plethora of mechanical stimuli arising from eye movements and changes in intraocular pressure. However, the mechanisms and pathophysiological outcomes underlying the intercellular crosstalk in the context of tissue mechanics in the CNS are poorly understood.

Of the many cell types in the CNS, microglia are particularly well-positioned to coordinate tissue organization. Originating from the yolk sac, microglia populate neural tissues early in development prior to the formation of blood vessels[9] and shape the neural circuits[10,11]. In adults, the key functions of microglia are tissue surveillance and defense, as microglia are the only CNS cell type that is constantly moving and encountering changes in the tissue microenvironment[12,13]. In this study, we focused on tissue microarchitecture and retinal vascularization, one of the best models of embryonic neural vascular development. The retina, characterized by the highest oxygen consumption per unit weight, is the tissue that most depends upon properly-patterned and precisely-timed vascularization. Consequently, it is not surprising that any vascular anomaly in the retina causes impaired vision[14]. For example, age-related macular degeneration (AMD) is a leading cause of vision loss and occurs in >20% of the population[15]. In AMD, neovascularization originates either from choroidal or retinal vasculature, with the latter pathology referred to as retinal angiomatous proliferation (RAP)[16,17] and its mechanism is not understood. The depth of retinal vasculature is conserved and vascularization is strictly limited to the inner layer of the retina, whereas its deeper layers of photoreceptors remain completely avascular to ensure normal vision[18]. This microarchitectural precision of retina suggests the presence of a tightly-regulated mechanism.

Since microglia are causatively connected to most inflammatory and neurodegenerative disorders from multiple sclerosis to Alzheimer's disease to depression[11,19,20], numerous studies have aimed to identify microglial genes that are differentially expressed in human pathologies. Interactions between microglia and its microenvironment or with other cells are often mediated by cell adhesion receptors called integrins. The main microglial integrin αmβ2 facilitates axon pruning and pathological microgliosis[10]. Nevertheless, cell adhesion receptors rarely appear amongst key targets in genome-wide searches[21]. In contrast, the short list of overlapping targets compiled from different studies contains several intracellular adapters, among which FERMT3 (also known as Kindlin3 (K3)) is hypomethylated and upregulated in multiple sclerosis[21]. Similarly, changes in Kindlin3 levels were observed in Alzheimer's and Parkinson's diseases, schizophrenia, HIV-associated neurocognitive disorders, and more dramatically in high-grade glioblastomas as identified from gene expression profiles from curated Gene Expression Omnibus (GEO) datasets of patients[22]. In the CNS, Kindlin3 is exclusively expressed in microglia;[23] however, it is not yet listed in any immune or neurological pathways. The main function of Kindlin3 is to bind and activate integrins[23]. In humans, Kindlin3 deficiency causes devastating bleeding and immune disorder, known as LAD-III, often accompanied by cerebrovascular complications[24]. It is

possible that Kindlin3 coordinates key microglial functions during tissue development and disease that may be either dependent on or independent of cell adhesion receptors.

In this study, we demonstrate that the pattern of vascular networks, positioned between the nuclear layers of increasing stiffness is controlled by microglia. Microglial bipolarization and proangiogenic TGFβ1 signaling are closely linked to the stiffness of the respective retinal layers. Increasing stiffness in hydrogels in vitro directly induces microglial polarization and a reduction in TGFβ1, which in turn is responsible for vascular restriction in vivo. Knockout of Kindlin3 in microglia impairs microglial stiffness sensing in vitro and dysregulates TGFβ1 in vitro and in vivo, leading to severe vascular abnormalities in the retina. This pathology is corrected by either microglial depletion or by microglia-specific knockout of TGFβ1. Mechanistically, this Kindlin3 function of regulating TGFβ1 is independent of microglial integrins and Kindlin-integrin interactions and is causatively connected with cellular contractility.

## Results

**Microglia migrate through stiffness gradient at development.** The retina has a highly conserved, layered structure with three vascular layers that are positioned exactly between the respective stiffer nuclear layers. As shown in Fig. 1a, while the deep vasculature is positioned on top of the outer nuclear layer (ONL), the intermediate vascular network grows between the outer and inner nuclear layers (INL). A stiffness map of the retina generated with atomic force microscopy (AFM) revealed that vascular networks are separated by the stiffer INL and ONL, characterized by elastic modulus of $2 \pm 0.2$ and $6 \pm 0.8$ kPa, respectively. At the same time, the stiffness of the vascularized outer plexiform layer (OPL) was $0.7 \pm 0.03$ kPa (Fig. 1a). Microscopy analysis showed that the stiffest ONL (6 kPa) had >3-fold higher nuclear density compared with the INL (2 kPa) (Fig. 1a), which likely contributes to its high stiffness.

During retinal development, microglia first populate the superficial layer and then migrate into the deeper layers, thereby experiencing significant changes in tissue stiffness (Fig. 1b and Supplementary Fig. 1a, b). As evident from the cross-sectional and lateral views of the retina shown in Fig. 1b and Fig. 1c, respectively, microglia are ramified within the softer OPL, but they become more widely spread and tightly wrapped around blood vessels on the stiffer ONL. Deep microglia on the ONL (6 kPa) are characterized by 2.4- and 4.5-fold changes in the vertical ramification index at P12 and P16, respectively, compared with P9, whereas intermediate microglia within the softer INL remained ramified (2 kPa) (Fig. 1d). Between P9 and P16, intermediate microglia form longer and more branched processes (Fig. 1e); however, they remained ramified with a cell polarity index between 1 and 2 (Fig. 1f). In contrast, deep microglia on the stiffest ONL showed time-dependent changes from ramified (polarity index $1.5 \pm 0.1$ at P9) to a bipolar rod shape (polarity index $3.6 \pm 0.3$ at P16). Together, these results show that during retinal vascular development microglia migrate through layers of increasing stiffness and undergo polarization by assuming a bipolar rod shape on the stiffest ONL.

To demonstrate whether tissue stiffness coordinates microglial polarization, we adapted a new method developed for the CNS that allows softening live tissues without damage[3]. Treatment of live retinas with chondroitin sulfate (CS) for 6 h decreased retinal stiffness from ~900 to 400 Pa based on AFM (Fig. 1g), while microglia remained not only viable with moving processes but also resting as judged by low expression of activation marker, CD68 (Fig. Supplementary Fig. 1c-e). Although CS might initiate

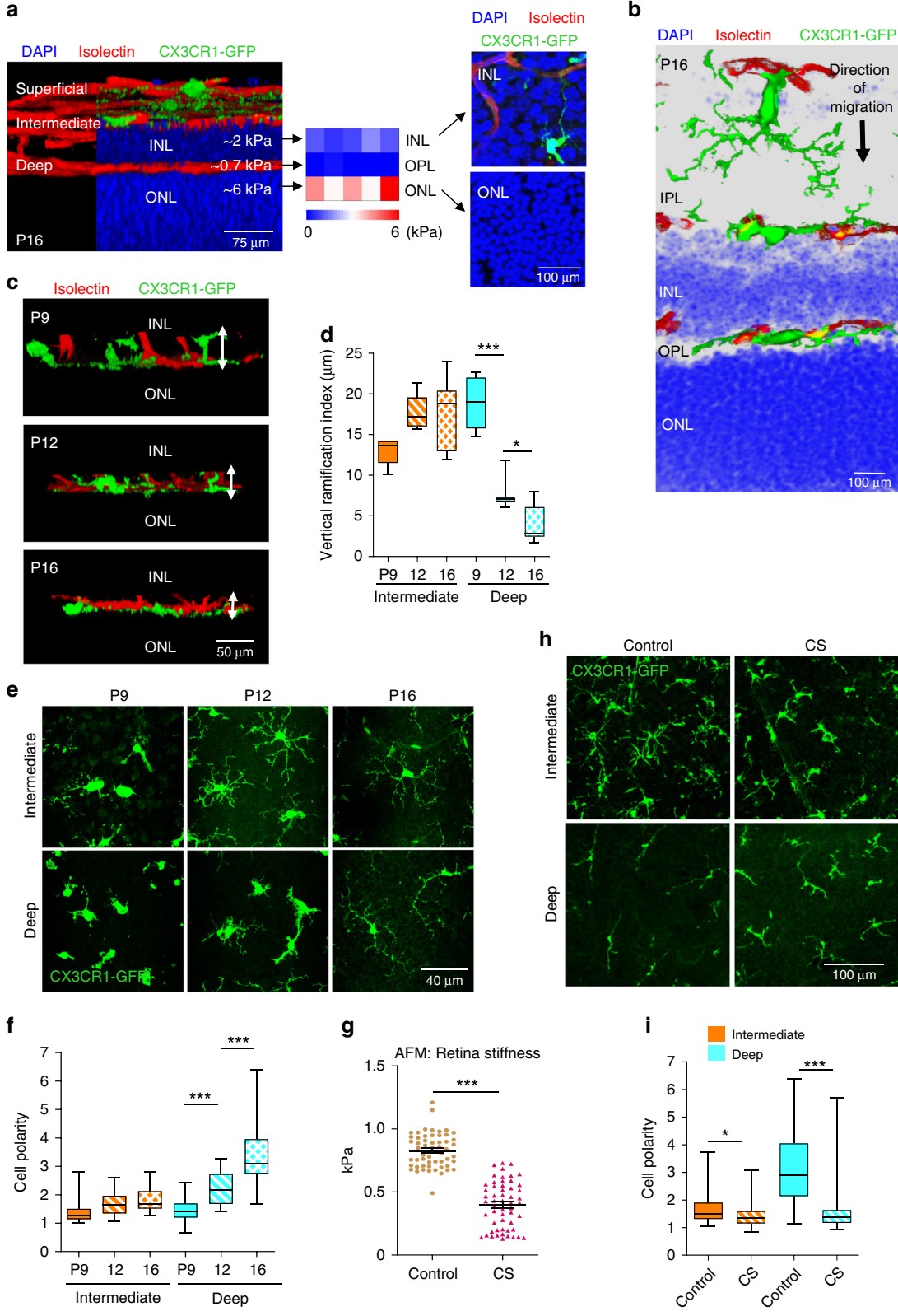

microglia activation[25], CD68 expression was not increased after 6 h treatment with CS in vivo (Supplementary Fig. 1e). Likewise, CS treatment in vitro did not significantly affect microglial morphology nor its activation status (Supplementary Fig. 1c, d). Together, these data indicate that treatment with CS for 6 h was sufficient to soften the retina without detectable microglia activation, which requires a substantially longer exposure[25]. CS treatment decreased the polarity index of deep microglia from ~3.5 to 1.7 and intermediate microglia from ~1.7 to 1.5 (Fig. 1h, i). The length of the major processes and the number of processes were also altered by CS treatment (Supplementary Fig. 1f, g). Thus, microglial polarization reflects increasing tissue stiffness not only during development, but also in this experimental model of tissue softening.

**Fig. 1 Microglia experience changes in tissue stiffness during development. a** A lateral view of 3D-reconstituted whole-mount P16 mouse retina with CX3CR1-GFP-expressing microglia (green) stained with isolectin to detect vasculature (red) and DAPI for nuclei (blue). The average stiffness of the INL, OPL, and ONL as measured by AFM are shown. Their stiffness map ($N = 3$ retinas) and the immuno-histochemical images representing the nuclear densities (blue) are shown on the right. **b** Cross-section of a P16 retina representing the direction of microglia migration from the ganglion cell layer toward the ONL. DAPI reveals the densities of nuclei in the INL and ONL. **c** 3D-reconstituted lateral view of whole-mount P9, P12, and P16 retinas. White arrows indicate the height of microglia expressing CX3XR1-GFP from the ONL. **d** Average vertical ramification index (cell height) of microglia from the intermediate and deep vascular plexi at retinal developmental ages of P9, P12, and P16. One-way ANOVA with Bonferroni's post hoc analyses; $P < 0.0001$ (***), $P = 0.0426$ (*); $N = 7$ cells from three mice each. **e** Age-dependent changes in microglia morphology and polarization. Images were taken at an approximate distance of 500–1500 μm from the optic nerve. **f** Quantitation of microglial cell polarity measured as a ratio of length to width. One-way ANOVA with Bonferroni's post hoc analyses; $P < 0.0001$ (***); $N = 35$ cells from five mice each. **g** AFM measurements of stiffness within a distance of 700–1200 μm from the optic nerve in P16 retinas that were treated with vehicle (control) or CS for 6 h (two-tailed $t$-test; $P < 0.0001$ (***), $N = 55$ measurements on four mice). Center line is mean and error bars represent standard error of mean (SEM). **h** Confocal microscopy image stacks of microglia in the intermediate and deep layers of control and CS-treated retinas. **i** Quantification of microglial polarization in deep and intermediate layers upon CS treatment of P16 retinas (one-tailed $t$-test; $P = 0.0372$ (*), $P < 0.0001$ (***); $N = 42$ cells from three mice each). Center line of box plots represents the median, bound of box shows 25th to 75th percentiles, and upper and lower bounds of whiskers represent the maximum and minimum values, respectively.

**Microglial response to stiffness is integrin-independent.** To substantiate the microglial responses to stiffness in vitro, we used hyaluronic acid-based hydrogels coated with fibronectin. Similar to our in vivo results, stiff hydrogels promoted bipolarization of primary microglia. An increase in substrate stiffness by 10-fold induced an ~3-fold increase in the polarity index of wild-type (WT) microglia (Fig. 2a, b). Knowing that bipolarization of microglia might trigger changes in secretory function, we focused on factors that are (a) exclusively produced by microglia in the CNS[26,27], and (b) known to be associated with changes in the CNS microenvironment during development, aging, and pathologies[28,29]. At the top of this list is TGFβ1, which is a key regulator of both microglia and the endothelium and is implicated in neurovascular pathogenesis by mechanisms that are yet to be established[30,31].

As shown in Fig. 2c, stiff substrates induced more than a 2-fold reduction in microglial TGFβ1 indicating mechanosensitive response. The same pattern was observed in vivo. TGFβ1 in retinas is expressed almost exclusively by microglia with ameboid non-polarized microglia expressing the higher levels compared with polarized cells (Supplementary Fig. 2a). Spatiotemporal analysis of TGFβ1 and its signaling in vivo further supported the existence of the mechanosensing mechanism observed in vitro. Microglial bipolarization on stiff ONL at P16 (Fig. 1f) was accompanied by a dramatic reduction in microglial TGFβ1 (Supplementary Fig. 2b, c). TGFβ1 can act through autocrine mechanism, therefore pSMAD3 in microglia reports on TGFβ1 signaling in these cells. As shown in Supplementary Fig. 2d-f and quantified in Fig. 2d, the ameboid/non-polarized intermediate microglia exhibited 3-fold higher pSMAD3 levels compared with deep polarized microglia.

Consequently, endothelial pSMAD3 levels closely followed TGFβ1 pattern in microglia. Microglia polarization from P12 to P16 triggered a 3-fold reduction in endothelial pSMAD3 within the deep vascular layer, while no significant change occurred in the intermediate layer where microglia remained ramified (Fig. 2e, f). This reduction in TGFβ1 signaling in the deep vasculature coincided with termination of further growth into the ONL (Fig. 1), thereby implicating microglial TGFβ1 in the restriction of excessive retinal vasculature. Moreover, microglia-specific knockout of TGFβ1 (CX3CR1-cre; TGFβ1$^{f/f}$ mice, Fig. 2g–i) resulted in ablation of pSMAD3 localization to endothelial cells (Fig. 2i), further confirming the key role of microglial TGFβ1 signaling within retinal layers of varied stiffness.

It is well-accepted that mechanotransduction depends upon integrins, and, possibly, upon integrin adaptors, including Kindlin3. Accordingly, we compared microglial responses within the stiff ONL using knockouts of the main microglial integrins, β2

(CD18 hypomorph) and β1 (CX3CR1-cre; β1$^{f/f}$), as well as in a microglia-specific Kindlin3 knockout (CX3CR1-cre;K3$^{f/f}$) and knock-ins expressing either low (K3KI) or normal (K3KI-flp) levels of mutant Kindlin3 that was unable to bind to or activate integrins (protein levels are shown in Supplementary Fig. 3a-c). Microglia lacking β2 and β1 integrins exhibited spreading and polarization in vitro that were similar to WT, whereas only Kindlin3-deficient microglia showed defective spreading and polarization (Supplementary Fig. 3d, e). In all of these lines, microglia populated the CNS normally and eventually underwent bipolarization on the stiff ONL (Fig. 2j). Surprisingly, most of the microglia either lacking individual integrins or expressing mutant Kindlin3 (K3KI-flp, which is integrin-binding defective) underwent detectable bipolarization on the ONL (Fig. 2j, k). Only the absence of Kindlin3 in CX3CR1-cre;K3$^{f/f}$ or low levels of Kindlin3 in K3KI completely impaired microglial bipolarization on the stiff ONL (Fig. 2j–l). The ability of microglia to respond to the stiff ONL directly correlated with expression levels of Kindlin3, but not with its ability to bind to and activate integrins (as in K3KI-flp; Fig. 2m).

Likewise, Kindlin3-deficient microglia completely failed to polarize on hyaluronic acid (HA, a major constituent of the extracellular matrix closely resembling the retinal microenvironment) hydrogels of increasing stiffness in vitro (Fig. 2n, o), and thereby continuously expressed high levels of TGFβ1 regardless of the substrate stiffness (Fig. 2p). Similar results were observed using silicone gels with a stiffness range similar to distinct retinal layers of 0.2, 0.5, and 2 kPa (Supplementary Fig. 4a, b).

**Kindlin3 is essential for microglial polarization on ONL.** To better understand microglial responses to changes in tissue stiffness during development, we performed lineage tracing of microglia in both WT and K3KI retinas between the ages of P9 and P16 using a CX3CR1-GFP reporter (Fig. 3a, b). Co-staining for markers Iba-1 and Tmem-119 demonstrated the microglial specificity of CX3CR1 as well as the exclusive presence of Kindlin3 in microglia of developing retinas (Supplementary Fig. 4c-e). Within the intermediate and deep layer, WT microglia matured and formed branched processes in close contact with the vasculature, whereas K3KI microglia exhibited delayed maturation with limited branching (Fig. 3a, c; Supplementary Fig. 5a-g). Within the deep ONL (P9-P16), WT microglia became bipolarized (polarity of deep microglia increased by >2-fold) and aligned along the blood vessels. In contrast, deep K3KI microglia exhibited no significant changes on the stiff ONL, remaining ramified and lacking alignment (Fig. 3b–d).

To understand the consequences of microglial abnormality in Kindlin3-deficient mice, we performed RNA microarray of entire

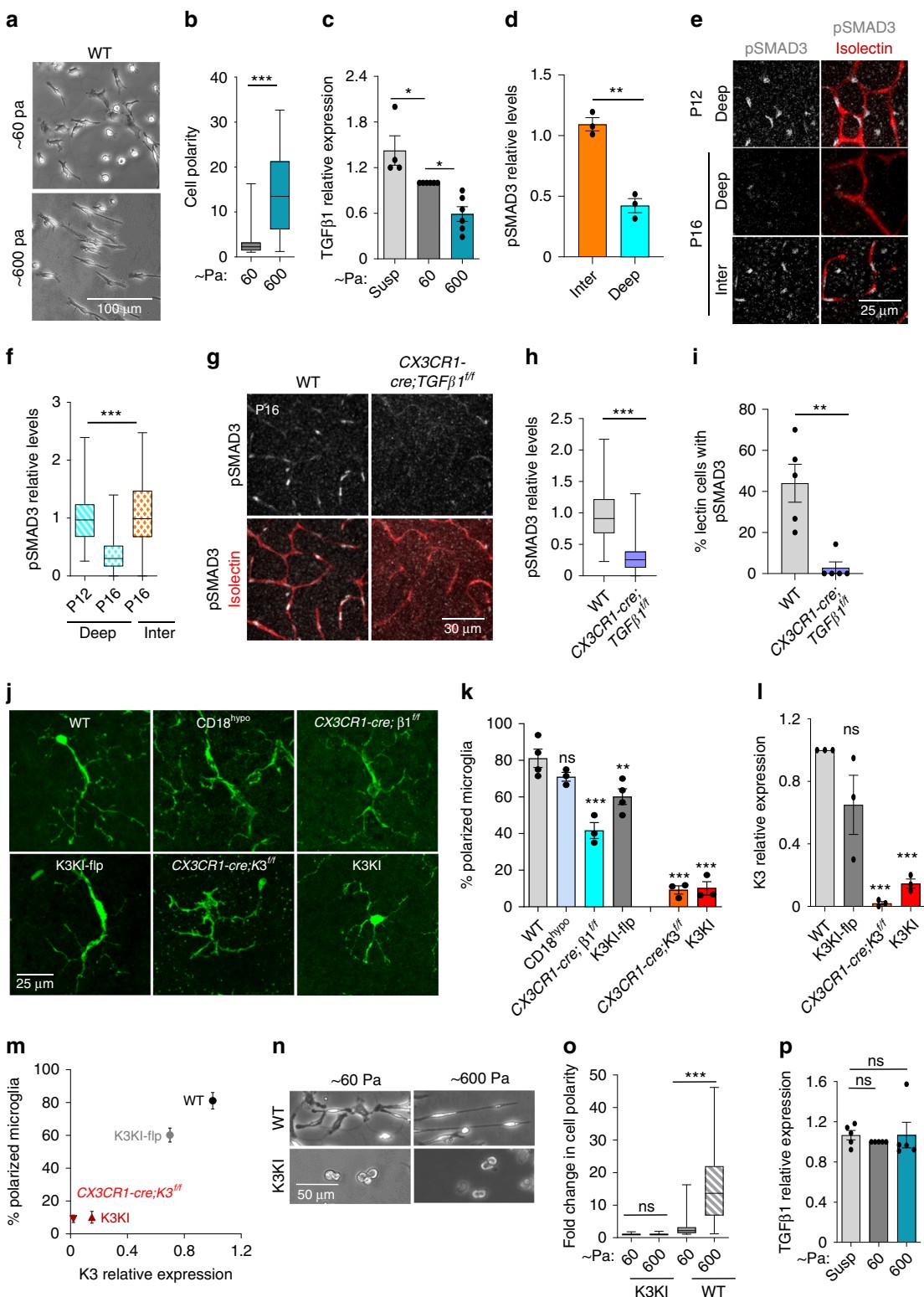

retinas from *CX3CR1-cre;K3f/f* and control mice, which revealed changes in ~140 genes, primarily reflecting the main cellular component of retinas, the photoreceptors (supplementary file). Several pathways were overrepresented, including sensory perception and eye development (Supplementary Fig. 6a). These results show that while the lack of Kindlin3 in microglia affected the general pathways of eye development and photoreceptor function, changes within microglia specifically were difficult to discern. Only analysis of primary isolated microglia by ELISAs combined with QPCR revealed abnormally high levels of TGFβ1, both at the RNA and protein levels, in Kindlin3-deficient microglia as compared with WT; there were no significant changes in TNFα, IL1β, or other key regulators of angiogenesis such as sFLT and VEGFA (Fig. 3e; Supplementary Figs. 6b and 7a-g). Furthermore, high levels of TGFβ1 produced by K3KI microglia and its stimulatory effect on endothelium was confirmed using monkey retinal endothelial RF6A cells as a reporter. The percentages of pSMAD3-positive RF6A cells were

**Fig. 2 ECM stiffness regulates TGFβ1 expression by microglia. a** Representative phase-contrast images of WT primary microglia spread on HYSTEM gels of ~600 Pa and ~60 Pa stiffness overnight from four independent experiments. **b** Bar graph quantifying the microglial cell polarity measured as a ratio of length to width (two-tailed $t$-test; $P < 0.0001$ (***); $N = 59$ cells from four experiments). **c** TGFβ1 mRNA expression measured by QPCR in WT microglia in suspension (Susp) and spread on HYSTEM gels of ~60 and ~600 Pa ($N = 4$ experiments). One-way ANOVA with Bonferroni's post hoc; $P_{sup-60} = 0.03$ (*) and $P_{60-600} = 0.02$ (*). **d** Relative pSMAD3 levels in microglia of intermediate and deep layer (two-tailed $t$-test; $P = 0.0011$ (**); $N = 3$ mice). **e** Staining of pSMAD3 in isolectin-positive endothelium of retinal vascular plexus from P12 and P16 WT mice. At P12, intermediate plexus is not yet established. **f** Quantification of pSMAD3 immunofluorescence intensity (two-tailed $t$-test; $P < 0.0001$ (***); $N = 102$ cells from three mice). **g** Representative confocal images of pSMAD3 staining in the vasculature (isolectin) of WT and TGFβ1 knockout (*CX3CR1-cre; TGFβ1f/f*) mice ($N = 4$ mice). **h, i** Bar graphs showing relative pSMAD3 immunofluorescence intensity (two-tailed $t$-test; $P < 0.0001$ (***); $N = 79$ cells from three mice) and percent endothelial cells (lectin stained) (two-tailed $t$-test; $P = 0.0028$ (**); $N = 5$ mice). **j** Representative confocal images of microglia on the ONL of retinas from P16 WT, β2 knockout (CD18hypo), β1-knockout (*CX3CR1-cre;β1f/f*), K3KI-flp, K3KI, and microglia-specific Kindlin3-knockout (*CX3CR1-cre;K3f/f*) mice. **k** Bar graph of percentage of microglia showing bipolar phenotype, i.e., cells with a ratio of length to width > 3 on the ONL of retinas from P16 mice (one-way ANOVA with Dunnett's post hoc comparing WT to other genotypes; $P_{CD18hypo} = 0.3278$ (ns), $P_{CX3CR1-cre;β1f/f} < 0.0001$ (***), $P_{K3KI-flp} = 0.0062$ (**), $P_{CX3CR1-cre;K3f/f} < 0.0001$ (***), $P_{K3KI} < 0.0001$ (***); $N = 4$ mice for WT and K3KI-flp, and 3 mice for all other groups. **l** QPCR analysis Kindlin3 expression in primary microglia isolated from Kindlin3-deficient (K3KI and *CX3CR1-cre;K3f/f*) mice in comparison to K3KI-flp and WT mice (one-way ANOVA with Dunnett's post hoc; $P_{K3KI-flp} = 0.0780$ (ns), $P_{CX3CR1-cre;K3f/f} = 0.0003$ (***), $P_{K3KI} = 0.0006$ (***); $N = 3$ experiments. **m** Correlation graph showing dependence of microglial polarization in retina on Kindlin3 expression levels in microglia ($N = 3$ mice). **n** Representative phase-contrast images from four experiments of WT and K3KI microglia on fibronectin-coated HYSTEM gels of ~600 and ~60 Pa stiffness. **o** Bar graph comparing fold changes in cell polarity of microglia spread on Hystem gels (two-tailed $t$-test; $P = 0.1905$, $P < 0.0001$ (***); $N = 60$ cells from four experiments). **p** QPCR analysis of TGFβ1 mRNA levels in K3KI microglia in suspension (Susp) and spread on HYSTEM gels of ~60 and ~600 Pa (one-way ANOVA with Dunnett's post hoc, $P_{Susp-60} = 0.7750$ (ns), $P_{Susp-600} = 0.9999$ (ns); $N = 5$ experiments). All bar graphs are represented as mean and error bars are SEM. Center line of box plots represents the median, bound of box shows 25th to 75th percentiles, and upper and lower bounds of whiskers represent the maximum and minimum values, respectively.

20 and 60% in the presence of WT and K3KI microglia, respectively (Supplementary Fig. 7h, i), demonstrating that TGFβ1 overexpressed by K3KI microglia dramatically augments canonical pSMAD3 signaling in endothelium. Finally, to confirm the requirement of Kindlin3 for microglia polarization in vivo, in support of our in vitro observations (Fig. 2n, o), K3KI retinas were treated with CS for 6 h. As shown in Fig. 3f, g, the CS treatment to decrease retinal tissue stiffness depolarized the WT microglia on the ONL, but it had no substantial effect on Kindlin3-deficient microglia.

**Microglia and Kindlin3-dependent retinal vascularization.** In the CNS, microglia are a main source of TGFβ1; therefore, high expression of TGFβ1 by K3KI microglia resulted in a 3.5-fold increase in endothelial pSMAD3 staining in vivo as compared with WT (Fig. 4a, b). Upregulation of pSMAD3 was confirmed by western blot analysis of retinal extracts (Supplementary Fig. 8a). Thus, this high level of proangiogenic TGFβ1 signaling substantially affected the retinal vasculature.

Indeed, while the vasculature in WT retinas had completed vertical growth and formed the final three layers at P16, blood vessels in K3KI were continuously growing into the normally avascular ONL of photoreceptors (Fig. 4c, d). Strong pSMAD3-staining was especially prominent in these endothelial neovascular sprouts, growing beyond the deep retinal layer in K3KI mice (Fig. 4e). Premature vertical neovascular growth in K3KI was also observed at P6, but not at earlier stages (Supplementary Fig. 8b-d). Depth labeling of the resulting vasculature (Fig. 4f) shows that in contrast to WT, K3KI vasculature is characterized by a number of vertical neovascular sprouts with an average length of 17.7 ± 0.9 µm, not only reaching into the typically avascular ONL, but even penetrating beyond it (Fig. 4f). By the age of P21, neovascular sprouts growing from the deep vascular plexus into the ONL were transformed into large bulbous blood vessels resembling vascular malformations. Some of these malformations merged together to form lesions under the photoreceptors, nearly resembling a forth vascular layer, which is never seen in normal mice (Fig. 4g). By P60, 15–20% of the Kindlin3-deficient mice retained these neovascular lesions (Supplementary Fig. 8e). "Skeletonization" of images (Fig. 4h) revealed that K3KI vasculature had 1.7-fold higher density of vascular loops, whereas

the loops themselves were 4.4-fold smaller as compared with WT (Fig. 4i). The angles of the connecting blood vessels between the vascular layers were conserved in WT retinas (75 ± 5.1°); however, this parameter was also abnormal in K3KI mice (23 ± 7.2°), resulting in nearly 1.8-fold higher density of vascular junctions compared with WT (Fig. 4j, k).

**Microglial integrins are dispensable for vascular architecture.** To show that vascular restriction depends on microglia, we analyzed a microglia-specific Kindlin3 Knockout (under *CX3CR1-cre*), which disables microglial polarization within the deep layer (Fig. 2j, k). This knockout caused abnormally dense (Fig. 5a, b) and misguided retinal vasculature growing into the ONL (Fig. 5c, d), which is a phenotype similar to K3KI mice. At the same time, disruption of Kindlin3-integrin binding alone in K3KI-flp microglia had no effect on microglial bipolarization (Fig. 2j, k), TGFβ1 expression (Fig. 5e), or vasculature (Fig. 5f). Likewise, vascular phenotype was absent in the CD18 hypomorphic mice lacking the main microglial integrin β2 (Fig. 5g). In microglia-specific β1-integrin knockout mice (*CX3CR1-cre; β1f/f*), the vascular density within the existing layers was somewhat higher than in controls; however, similar to β2-knockout mice, the main phenotype of excessive sprouting beyond the ONL was absent (Fig. 5h and Supplementary Fig. 9a, b). A quantitative comparison of Kindlin3 and integrin knockout mice shows that neovascular sprouts characteristic of retinal pigment epithelium (RPE) were caused by the lack of microglial Kindlin3, independent of its ability to bind to integrins (Fig. 5i).

**Rescue of abnormal vasculature by microglial TGFβ1 depletion.** Depletion of microglia at the early postnatal stages of retinal development delays vascularization (Supplementary Fig. 9c). However, the same depletion procedure performed after P14, when vascular layers are nearly formed, slightly increases vascular density (Supplementary Fig. 9d). Thus, in agreement with previous reports[32,33], while retinal microglia are angiogenic during earlier stages (before P12–14), it seems to restrict the vasculature at later stages[34]. Thus, the timing of microglial depletion is critical. To show the causative role of microglia in vascular abnormalities of K3KI mice, we depleted

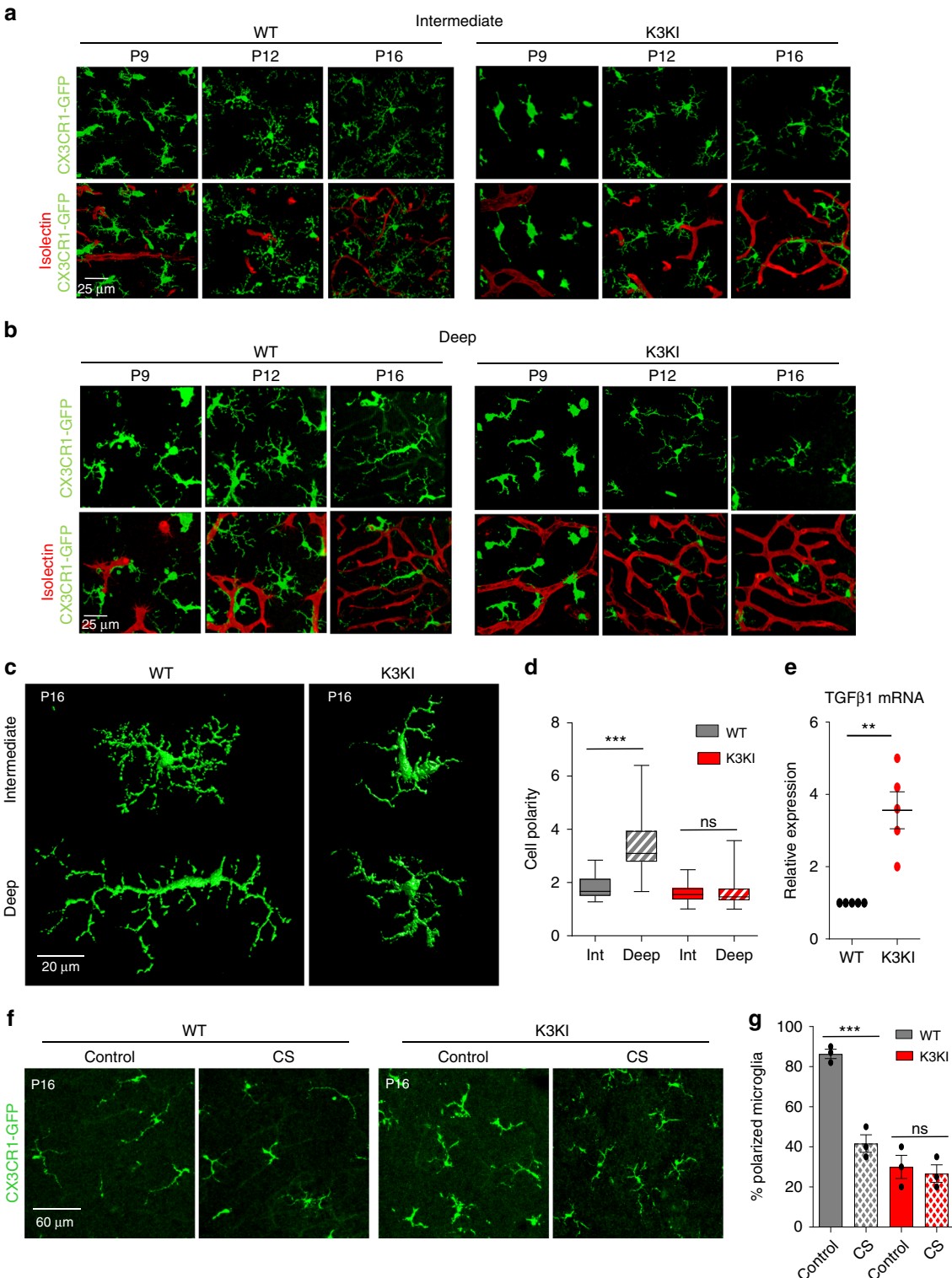

>95% of microglia with pexidartinib (Fig. 6a, b) before the vascular layers were completed. Microglial depletion normalized vascular density in K3KI retinas to levels observed in WT retinas (Fig. 6a, c). Next, to ultimately prove that it is microglial TGFβ1 overexpression that caused misguided vasculature in *CX3CR1-cre;K3^{f/f}* and K3KI mice, we generated a double knockout of Kindlin3 and TGFβ1 in microglia, *CX3CR1-cre; K3^{f/f}/TGFβ1^{f/f}* mice. Downregulation of TGFβ1 signaling in endothelium of *CX3CR1-cre;K3^{f/f}/TGFβ1^{f/f}* mice was confirmed by dramatically reduced pSMAD3 staining (Fig. 6d, e) and western blotting for TGFβ1 (Supplementary Fig. 9e). The reduction in TGFβ1 led to normalization of dense vasculature in *CX3CR1-cre;K3^{f/f}/TGFβ1^{f/f}* retinas to levels observed in *CX3CR1-cre;K3^{f/f}/TGFβ1^{+/+}* control mice (Fig. 6d, f). Inducible knockout of TGFβ1 in microglia (*CX3CR1-cre;K3^{f/f}/TGFβ1^{f/f}*) did not affect microglial polarization or numbers (Supplementary Fig. 9f,g) but completely rescued the misguided vascular sprouting caused by Kindlin3 deficiency (Fig. 6f–h). Thus, TGFβ1 overexpression by Kindlin3-deficient microglia caused excessively dense and misguided vascularization.

**Fig. 3 Kindlin3 deficiency precludes microglial polarization on stiff substrates in vivo. a, b** Representative images of CX3CR1-GFP-expressing microglia within intermediate and deep vascular layers from P9-P16 isolectin-stained (red) whole-mount retinas from three or more mice. Note; polarization of deep WT but not K3KI microglia at P16, while intermediate microglia remained ramified. **c** 3D-reconstituted images of CX3CR1-GFP WT and K3KI microglia in P16 retinas. Representative of six or more mice per genotype. **d** Cell polarity measured as the ratio of cell length by width for intermediate and deep microglia in the retina of WT and K3KI P16 mice. One-way ANOVA with Tukey's post hoc showed significant microglial polarization by WT microglia in deep layers, but not by K3KI. $P_{WT\ int-deep}$ = < 0.0001 (***), $P_{K3KI\ int-deep}$ = 0.9999 (ns); N = 37 cells from six mice. **e** QPCR analysis for microglia-specific cytokine expression showed significantly higher expression levels of TGFβ1 mRNA in K3KI relative to WT primary microglia in culture (one-tailed t-test; P = 0.0037 (**); N = 5 experiments). **f** Confocal microscopy image stacks of deep-layer microglia in CS-treated WT and K3KI P16 retinas. WT microglia assumed a less polarized state in response to a decrease in retinal tissue stiffness (CS treated), while K3KI microglia showed no response in CS-treated retinas. N = 4 mice. **g** Bar graph representing a significant decrease in the percentage of polarized microglia only in WT retinas, but not in K3KI retinas, upon treatment with CS (two-tailed t-test; P = 0.0009 (***), P = 0.6702 (ns); N = 3 mice). All bar graphs and dot plots are represented as mean and error bars are SEM. Center line of box plots represents the median, bound of box shows 25th to 75th percentiles, and upper and lower bounds of whiskers represent the maximum and minimum values, respectively.

**Kindlin3 regulates TGFβ1 via actomyosin contractility and ERK.** Next we aimed to delineate the connection between Kindlin3, which controls microglial spreading and polarization, and TGFβ1 expression. In most cells, cell spreading is counteracted by the contractility of cytoskeleton (i.e., myosin contractility measured by myosin light chain phosphorylation (pMLC))[35]. In WT microglia, pMLC staining was low and localized close to the plasma membrane and actin cortex (Fig. 7a). The lack of Kindlin3 not only abolished pMLC co-localization with the membrane, but also resulted in a dramatic increase in pMLC levels, indicating higher cell contractility (Fig. 7a; Supplementary Fig. 10a). To causatively connect microglial polarization and TGFβ1 expression, we lowered cytoskeletal contractility using the myosin-II inhibitor blebbistatin[36] (Fig. 7b). Blebbistatin was able to "rescue" the spreading and polarization defects in K3KI microglia by increasing cell area by 4-fold (Fig. 7b). This "forced" polarization of K3KI microglia also normalized TGFβ1 expression to the levels observed in WT cells (Fig. 7c), demonstrating that microglial polarization directly controls TGFβ1.

Since high pMLC reflects high ERK-activity[37], we show that the lack of Kindlin3 disrupted ERK localization to the membrane and caused increased levels of pERK (Fig. 7d, e). Higher levels of pERK were also detected in K3KI retinas as compared with WT in vivo (Supplementary Fig. 10b). ERK might directly control TGFβ1 expression in phagocytic cells[38]; therefore, we tested whether pERK is responsible for high TGFβ1 expression in microglia. Inhibition of ERK reduced TGFβ1 expression as well as pMLC levels in microglia (Fig. 7f). This mechanism was confirmed using two independent CRISPR-K3KO clones of myeloid RAW cells (K3KO1 and 2), which mirrored K3KO and K3KI microglia in the lack of spreading (Fig. 7g), high levels of TGFβ1, and high contractility associated with high pMLC and pERK levels (Fig. 7h; Supplementary Fig. 10c). Similar to K3KI microglia, blebbistatin treatment of these K3KO cells reversed their hyper contractility, resulting in downregulation/normalization of TGFβ1 mRNA levels as analyzed by QPCR (Supplementary Fig. 10d-f). Inhibition of ERK, but not the P38 MAPK pathway, resulted in normalization of TGFβ1 in K3KO cells to levels observed in WT cells (Fig. 7i). Together, these results show that Kindlin3-mediated cell polarization controls TGFβ1 levels in myeloid cells in an ERK-dependent manner. Microglial bipolarization in stiff tissues in vivo and in vitro requires Kindlin3, but not integrins, which regulates vascular architecture via spatiotemporal control of TGFβ1.

## Discussion

In the CNS, surveying microglia constantly encounter mechanically heterogeneous regions of varying cellular density and matrix composition[3,39]. This heterogeneity is often augmented in various pathologies[6,12], especially those associated with microgliosis[40].

In this study, we demonstrate that during retinal development, microglia migrate through the mechanically heterogeneous tissue layers towards the stiffer ONL. This ability of microglia, described as durotaxis[12], suggests the presence of a mechanosensory mechanism. The retina represents an excellent model of tissue stiffness gradient. Since microglia populate certain layers at the precise times, the changes in microglia can be monitored in spatiotemporal manner, i.e., within the different layers in retinas of various ages. Within the stiffest ONL, microglia undergo a characteristic bipolarization, often observed in pathologies and known to be accompanied by changes in secretory function[41]. Treatment and softening of retinas with CS reduced polarization of WT microglia, but not Kindlin3-deficient microglia, thereby suggesting a connection between tissue stiffness and microglia polarization, which was further substantiated in vitro. By mimicking retinal stiffness in hydrogels, we demonstrated that bipolarization of microglia occurs on stiff substrates and leads to a dramatic reduction in TGFβ1, which is a potent regulator of cell proliferation and extracellular matrix composition[42] almost exclusively produced by microglia in the CNS. TGFβ1 signaling in retinas followed a similar layered pattern, showing a substantial reduction in polarized microglia on the stiffest ONL. Surprisingly, this microglial response requires the presence of Kindlin3, but not Kindlin3 binding to its main mechanosensory receptors, integrins. Microglia lacking Kindlin3 (CX3CR1-cre; K3$^{f/f}$ and K3KI) were not able to respond to the changes in stiffness in vitro, and lacked polarization in vivo, which are in contrast to cells expressing Kindlin3 defective in integrin binding (K3KI-flp). As a result, Kindlin3-deficient microglia overexpress TGFβ1 in vitro and in vivo, leading to high pSMAD3 levels in retinal endothelium and transformation of the vasculature into an overly dense and chaotic pattern. High TGFβ1 levels promote vascularization of normally avascular ONL of photoreceptors, a condition known as retinal angiomatous proliferation (RAP), which is a subset of age-related macular degeneration (AMD). The vascular lesions replace photoreceptors and might promote retinal detachment and vascular rupture. Importantly, in Kindlin3-deficient humans with severe bleeding, this pathology might be further exacerbated. This aberrant vascularity and excessive growth in Kindlin3-deficient mice was corrected by microglial depletion as well as by inducible knockout of TGFβ1 in microglia. In these experiments, both Kindlin3 and TGFβ1 excision were induced simultaneously at P1–5. This did not affect the presence of microglia in retinas, but substantially reduced TGFβ1 levels and SMAD signaling in endothelium, thereby normalizing vascular pattern. It appears that Kindlin3 knockout microglia produced a substantial amount of TGFβ1 prior to its excision, sufficient to support the formation of vascular layers. Besides microglia, other myeloid cells can potentially contribute to the observed vascular phenotype[43,44]. In

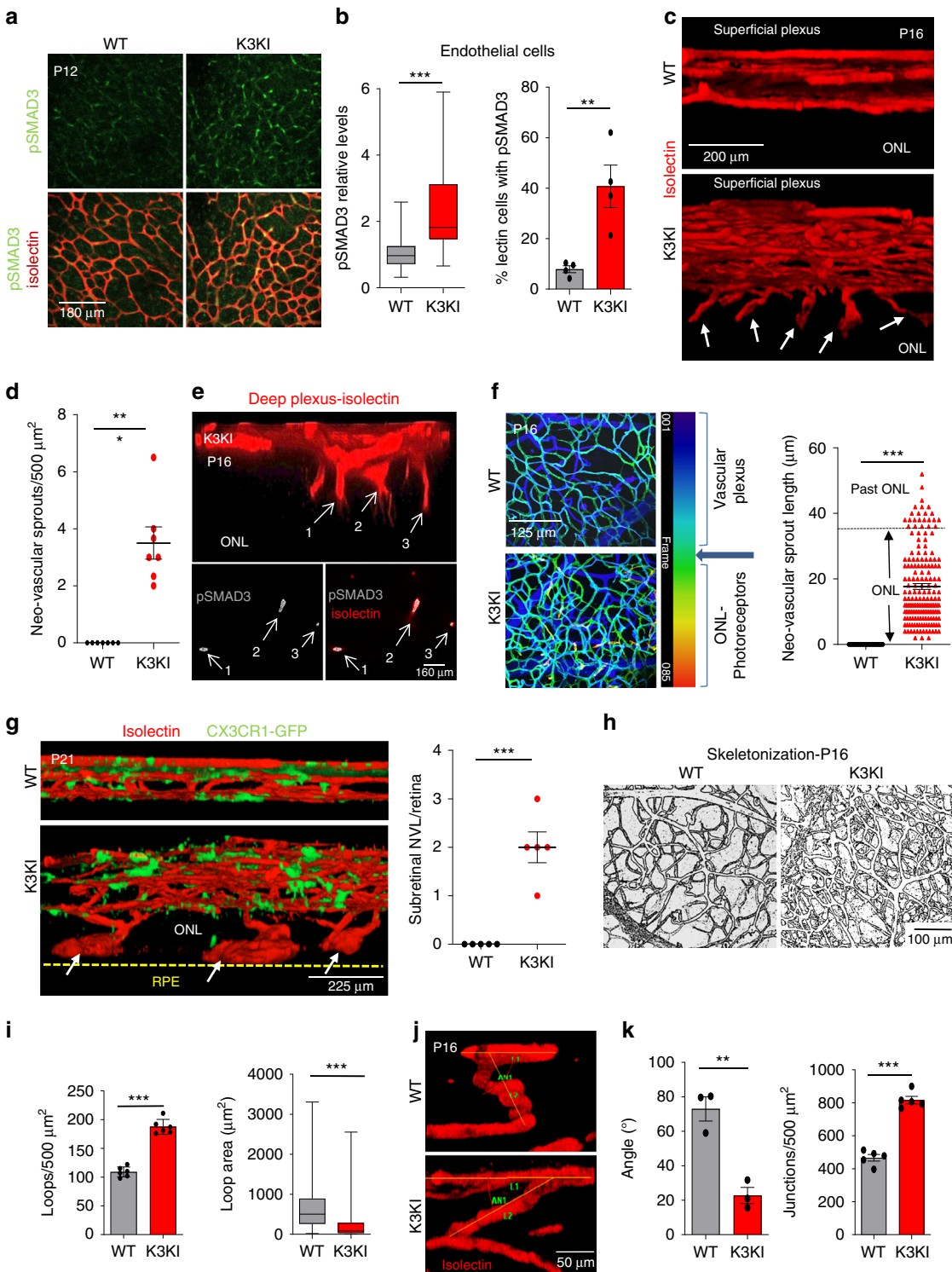

healthy developing retina these cells are represented by perivascular macrophages, however, their numbers are negligible (<15 cells/retina) compared with microglia at the same age[45], therefore, their contribution to the robust vascular phenotype is unlikely. Thus, this TGFβ1-dependent paracrine restriction of retinal vasculature is mediated by microglia and occurs at the stiffest ONL, a mechanical boundary generally known to dictate collective tissue organization[46]. TGFβ1 and its signaling are associated with vascular pathologies, rather than development serving as a driver of endothelial-to-mesenchymal transition[47]. In the CNS, TGFβ1 mediates directed growth, differentiation,

and anatomical orientation of axons[48]. Moreover, TGFβ1 might exhibit an autocrine effect on microglia, and as a result promote plasticity and differentiation[31]. Thus, this function of microglia is likely to interconnect tissue composition and mechanics with the overall neural tissue architecture and function.

Mechanistically, using the Kindlin3 specific knock-in mutant (K3KI-Flp) and individual integrin knockouts, we show that microglial function in vascular patterning, including responsiveness of microglia to substrate stiffness in vitro, depends upon Kindlin3, but not on its interaction with integrins. Thus, while Kindlin undoubtedly functions as an integrin adaptor and

**Fig. 4 Excessive and misguided retinal vasculature in Kindlin3-deficient mice. a** Deep vascular plexus of P12 retinas immunostained for pSMAD3 and isolectin. **b** Bar graphs showing relative pSMAD3 immunofluorescence intensity (two-tailed $t$-test; $P < 0.0001$ (***); $N = 60$ cells from three mice) and percent endothelial cells (lectin stained) with high pSMAD3 levels (two-tailed $t$-test; $P < 0.0084$ (**); $N = 4$ mice). **c** A lateral view of 3D-reconstituted vasculature from P16 retinas showing abnormal and excessive neovascular sprouts (indicated by white arrows) penetrating into the ONL. **d** Number of neovascular sprouts invading the ONL/500 μm$^2$ (two-tailed $t$-test; $P < 0.0001$ (***); $N = 7$ mice). **e** An enlarged view of the neovascular sprouts invading the ONL in K3KI P16 retinas showing pSMAD3 (arrows). The top panel shows a lateral view and the bottom panels are superimposed Z-planes corresponding to the vertical neovascular sprouts ($N = 3$ mice). **f** Neovascular invasion shown by depth color-coding of flat-mounted P16 retinas. Bar graph showing the length of neovascular sprouts penetrating the ONL, with some extending past the ONL (two-tailed $t$-test; $P < 0.0001$ (***); $N = 175$ neovascular sprouts from 15 mice). **g** A lateral view of 3D-reconstituted P21 retinas stained with isolectin. The K3KI retina shows sub-retinal neovascular lesions (NVLs) (white arrowheads) originating from the deep vascular plexus and resting on the RPE layer (indicated by the yellow line). Number of NVL per retina is shown in the graph on the right (two-tailed $t$-test; $P < 0.0001$ (***); $N = 5$ mice). **h** Skeletonized representation of vasculature in P16 retinas. **i** Quantitation of vascular loop density ($N = 6$ mice) and loop areas ($N = 125$ loops from six mice) of P16 retinas. Two-tailed $t$-test; $P < 0.0001$ (***). **j** A representative image from three mice showing the angle of vascular sprouts from the superficial vascular plexus in P16 retinas. **k** The bar graphs show the average vascular sprouts ($N = 3$ mice, two-tailed $t$-test; $P = 0.004$ (**)) and the number of vascular junctions within superficial vascular plexus of P16 retinas ($N = 5$ mice, two-tailed $t$-test; $P < 0.0001$ (***)). All bar graphs and dot plots are represented as mean and error bars are SEM. Center line of box plots represents the median, bound of box shows 25th to 75th percentiles, and upper and lower bounds of whiskers represent the maximum and minimum values, respectively.

activator, its role might extend beyond integrins. Besides integrins, Kindlin3 binds to the plasma membrane phospholipid, PIP2, through its main PH domain, and possibly through its second pseudo PH domain[49,50]. Kindlin3 contains three FERM domains and interacts with cytoskeletal proteins[51]. In the absence of Kindlin3, the cytoskeletal network is dissociated from the membrane, resulting in characteristic contraction of myosin in the center of the cell (Fig. 8). A similar contractile phenotype was observed in armadillo mutants, where the lack of apical adherent junctions caused myosin-II mislocalization and formation of a contracted ball at the center of the cell, precluding the cell shape changes necessary for gastrulation[52]. Likewise, the lack of Kindlin3 in microglia resulted in a highly contractile phenotype, possibly preventing microglia from bipolarization and alignment on rigid substrates. Kindlin3-deficient cells are characterized by a highly contractile actomyosin core, implicating Kindlin3 in actomyosin organization. We demonstrate that Kindlin-mediated microglia response to stiffness in vitro is dependent upon cell contractility and the phenotype can be rescued by loosening of myosin cytoskeleton. Actomyosin bundles are known to mediate mechanotransduction and serve as a scaffold for activation of ERK signaling, which is phosphorylated exclusively on actomyosin bundles in a myosin-II-dependent manner[53]. In phagocytic cells, ERK phosphorylation results in expression of TGFβ1[38] as observed in K3KI microglia. The entire chain of events leading to TGFβ1 overexpression was rescued in vitro by interference with cytoskeletal connections and ERK over-activation, indicating an importance of Kindlin3 link to cytoskeleton. At the same time, Kindlin3 binding to the membrane is clearly crucial for Kindlin3 functions, since PH domain deletion generally phenocopies Kindlin3 knockout[54].

Surprisingly, microglial migration in vivo and microglial populations in brain and retina during development are not affected by either by Kindlin3 or integrin deficiencies, which is consistent with the concept that microglia use ameboid rather than mesenchymal migration in vivo, similar to dendritic cells[55]. In contrast to in vivo migration, Kindlin3 is required for the maintenance of microglial cell shape and plasticity. The lack of Kindlin3 impaired formation of microglial processes and caused a complete lack of polarization and alignment. Microglial morphology changes with location, differentiation, and activation, and is reciprocally connected to its function[56,57]. In humans, changes in microglial shape serve as indicators of specific pathological processes in the CNS. The appearance of bipolar rod microglia is a hallmark of activation in many CNS diseases[58]. In glaucoma, bipolar rod microglia are associated with neurodegeneration and ocular hypertension[59]. We

demonstrate that microglial transition to a bipolar rod shape is triggered by tissue stiffness and serves as a necessary prerequisite for vascular restriction during development. However, the very same process may play a different role in pathologies associated with changes in substrate stiffness. Multiple examples of the crosstalk between biochemical and mechanical signals in the CNS support this notion[12,60].

## Methods

**Animals**. The Kindlin3 mutant knock-in mice (K3KI) were generated with the Q597W598 to AA mutation[61]. Briefly, a partial mouse *Kindlin3* gene in a BAC construct was introduced into a cloning vector. The mutation CAATGG to GCCGCC was introduced into exon 14 of the Kindlin3 gene by site-directed mutagenesis, giving rise to the mutant Kindlin3 protein with Q597W598 to AA substitution. A neomycin cassette was inserted into intron 13 of the Kindlin3 gene and the vector was electroporated into ES cells. Neo-positive ES cells were injected into C57BL6 blastocysts and implanted into pseudo-pregnant females. The chimeric male founders were crossed to WT C57BL/6 mice and the progeny positive for germline transmission were bred to generate K3KI mice. The Neo cassette present in the Kindlin3 gene resulted in very low levels of mutant Kindlin3 expression (Fig. 2l and Supplementary Fig. 3a). To remove the FRT-flanked neomycin resistance gene, K3KI mice were crossed with FLP1-expressing mice (Jackson laboratory) for four generations to obtain homozygous mice for FLP1 genes (K3KI-flp). Removal of the Neo cassette increased the expression levels of mutant Kindlin3 to levels similar to endogenous Kindlin3 of WT controls (Fig. 2l and Supplementary Fig. 3a). *CX3CR1*$^{GFP/GFP}$ mice expressing EGFP under *CX3CR1* promoter from the Jackson Laboratory were crossed to WT and K3KI mice. *CX3CR1*$^{GFP/-}$;WT/K3KI progeny were used for experiments. Microglia-specific inducible Kindlin3 knockout (*CX3CR1-cre;K3*$^{f/f}$) mice were generated by crossing Kindlin3 floxed (*K3*$^{f/f}$) mice with *CX3CR1-cre* (inducible) mice[23]. Briefly, Kindlin3 floxed mice were generated by flanking exon 2 of the *Kindlin3* gene with loxP sites on both sides. A neomycin cassette was inserted into intron 2. Neo-positive ES cells were injected into blastocysts and implanted into pseudo-pregnant females. Specific integration of targeted DNA fragments was verified by southern blotting and quantitative PCR with a probe for the inserted Neo gene. *CX3CR1-Cre* (tamoxifen-inducible) mice were obtained from Jackson Laboratory.

The *CX3CR1-cre;K3*$^{f/f}$ mice were bred with *TGFβ1*$^{f/f}$ (Jackson Laboratory) mice to obtain microglia-specific inducible Kindlin3 and TGFβ1 double knockout mice (*CX3CR1-cre;K3*$^{f/f}$/*TGFβ1*$^{f/f}$). The TGFβ1 knockout (*CX3CR1-cre;TGFβ1*$^{f/f}$) mice were derived by crossing the *CX3CR1-Cre* (tamoxifen-inducible) mice with the *TGFβ1*$^{f/f}$ mice. Integrin β1 knockout mice were generated by crossing *β1*$^{f/f}$ mice (Jackson Laboratory) with CX3CR1-Cre (tamoxifen-inducible) mice. Tamoxifen was dissolved in ethanol and then diluted in corn oil to a final concentration of 20 mg/ml. Tamoxifen was administered to mice orally from the age of P1–P5 at 4 μl/g of body weight. Integrin CD18 global hypomorphic mice were obtained from Jackson Laboratory. Pexidartinib (PLX3397) was dissolved in vehicle: 5% DMSO, 5% PEG 300, 5% Tween, and ddH2O to a stock concentration of 10 mg/ml. Pups were fed with pexidarininb (40 mg/kg) orally once daily for at least 7 days before terminating for experiments. The mice were housed in the Cleveland Clinic's Biological Resources Unit (BRU). The mouse room was maintained at 68–79 °F with 30–70% humidity and 14 h light/10 h dark cycle.

**Microglial isolation and culture**. Primary microglia were isolated from brains of postnatal day 1 (P1) mice. The cerebral hemispheres were carefully separated and

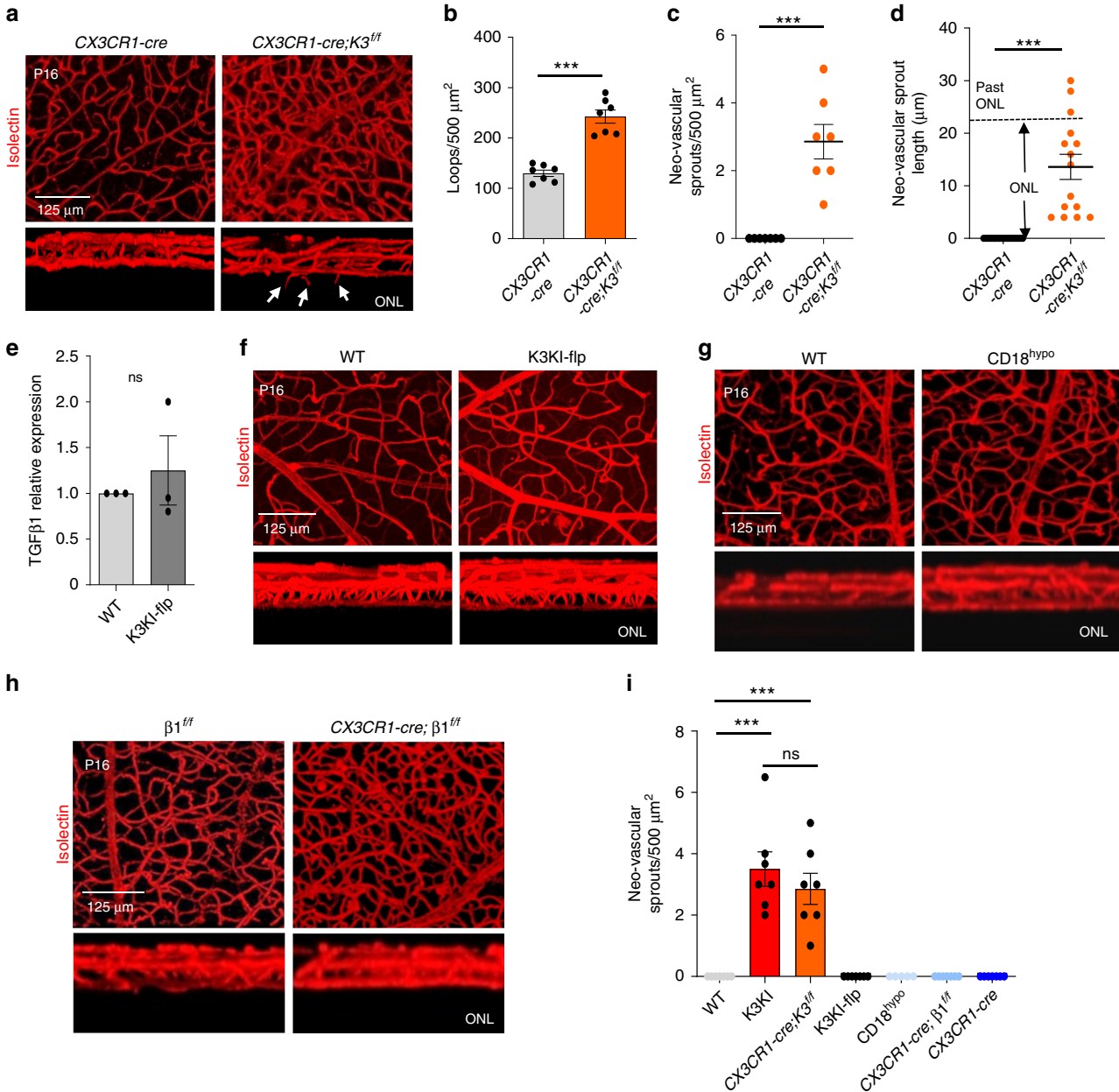

**Fig. 5 Kindlin3 levels are crucial for microglial function and normal vascularization. a** Excision of Kindlin3 in microglia is sufficient to induce vascular overgrowth. 3D-reconstituted P16 vasculature plexus from WT and microglia-specific conditional Kindlin3 knockout mouse (*CX3CR1-cre;K3^{f/f}*) retinas. Superimposed and cross-sectional views of the vascular layers are shown. The white arrowheads indicate the neovascular sprouts invading the ONL. Representative of three mice per genotype. **b–d** Quantification of deep vascular loop density (two-tailed *t*-test; *P* < 0.0001 (***); *N* = 7 mice), the average number of neovascular sprouts from the deep plexus per retina (two-tailed *t*-test; *P* < 0.0001 (***); *N* = 7 mice), and their respective lengths invading into the ONL (two-tailed; *P* < 0.0001 (***); *N* = 15 neovascular sprouts). **e** Mean relative expression of TGFβ1 mRNA as assessed by QPCR in K3KI-flp and WT primary microglial cells in culture (two-tailed paired *t*-test; *P* = 0.5440 (ns); *N* = 3 experiments). **f** A 3D reconstruction of retinal vascular plexus from whole-mount retinas of WT and K3KI-flp P16 mice stained with isolectin. A representative of four (K3KI-flp) and seven (WT) mice are shown. The top panel is a superimposed view of all three layers of vasculature and the bottom panel is a lateral view of the vascular plexus. **g** Retinal vascular plexus from whole-mount isolectin-stained retinas from control (WT) and β2 knockout (CD18^hypo) P16 mice. A representative of three (CD18^hypo) and seven (WT) mice is shown. **h** 3D reconstruction of retinal vascular plexus from whole-mount retinas from β1 knockout (*CX3CR1-cre; β1^{f/f}*) and control (β1^{f/f}) P16 mice stained with isolectin. A representative of four (β1^{f/f}) and seven (*CX3CR1-cre; β1^{f/f}*) mice are shown. **i** Quantification of the number of neovascular sprouts of the deep vascular plexus into the ONL of flat-mounted P16 retinas (one-way ANOVA with Tukey's post hoc; *P* < 0.0001 (***), *P* = 0.9205 (ns); *N* = 5 mice for *CX3CR1-cre;β1^{f/f}* and 7 mice for all other groups). All bar graphs and dot plots are represented as mean and error bars are SEM.

the meninges were removed. The tissue was dissociated in cold PBS into a slurry, passed through a 70 μm sieve, and centrifuged to pellet the cells. The cell pellet was then resuspended in DMEM/F12 (with 20% FBS, 100 U/ml penicillin and strep-tomycin, 0.25 μg/ml amphotericin B and supplementation of non-essential amino acids (NEAA)) and plated onto plastic dishes. The cells were cultured for 2–4 weeks before harvesting the pure floating microglia from the conditioned media by centrifugation at 700 × *g* for 10 min[23]. Microglia carrying the tamoxifen-inducible CX3CR1-cre with *K3^{f/f}* gene were treated with 10 nm 4-hydroxytamoxifen (4-HT) by addition to the medium every 48 h for efficient excision of the *Kindlin3* gene. For microglia activation assay, microglia were plated on coverslips and treated with 20 mg/ml CS and 10 ng/ml C5a or vehicle alone (control) for 6 h.

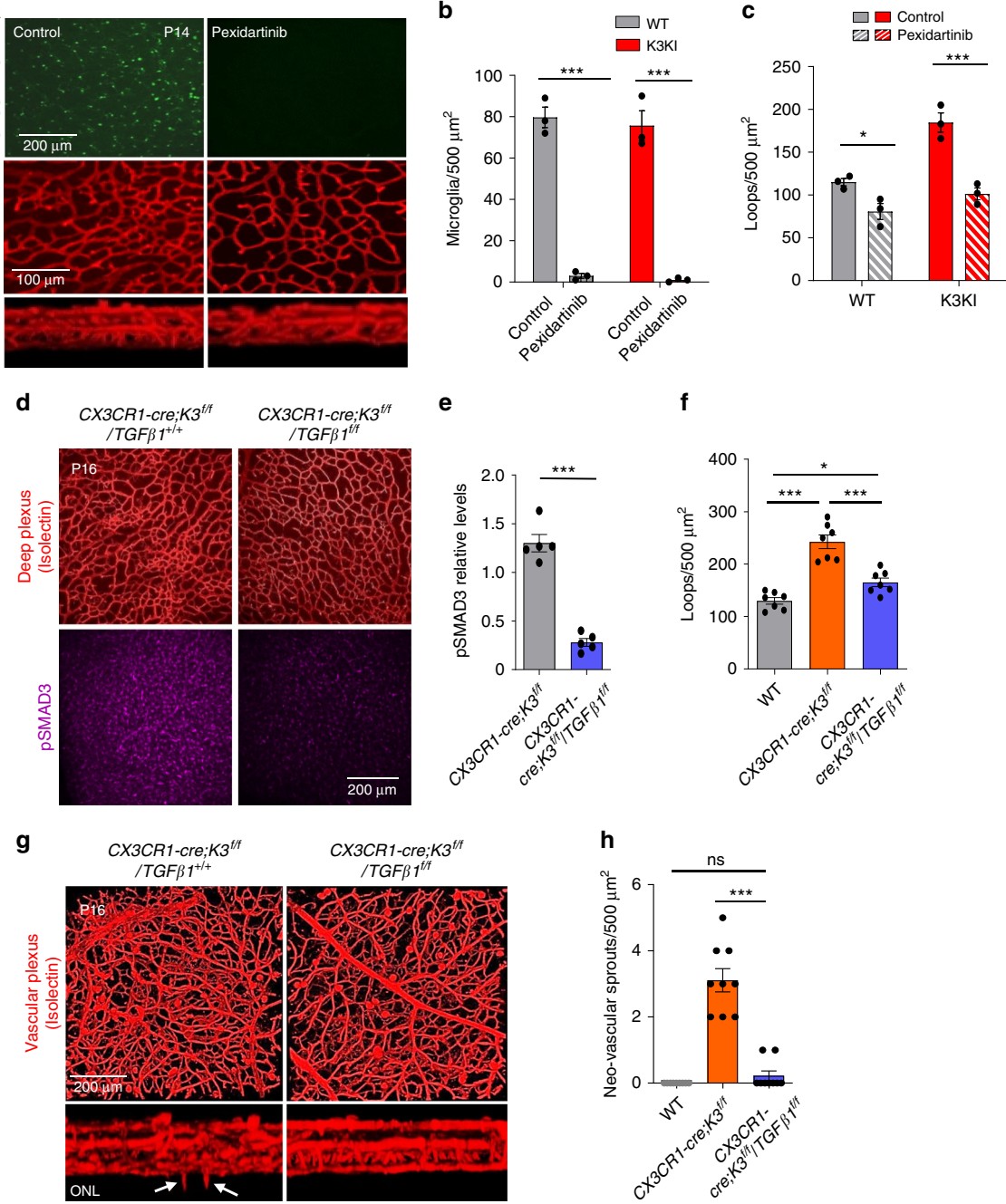

**Fig. 6 TGFβ1 knockout rescues Kindlin3 deficiency-induced abnormal vascular phenotype. a** Microglial depletion rescues vascular defects in K3KI retinas. P14 whole-mount retinas of K3KI mice treated with pexidartinib or vehicle alone (control). CX3CR1-GFP-expressing microglia (green) and vasculature stained with isolectin (red). A representative image of three experiments is shown. **b**, **c** Quantification of microglia numbers ($P < 0.0001$ (***)) and deep vascular loop density ($P_{WT} = 0.0401$ (*), $P_{K3KI} = 0.0002$ (***)) upon pexidartinib treatment (two-way ANOVA; $N = 3$ mice). **d** Rescue of abnormal vascular phenotype in P16 Kindlin3/TGFβ1 conditional double knockout (*CX3CR1-cre;K3$^{f/f}$/TGFβ1$^{f/f}$*) mice compared with their littermate controls; Kindlin3 conditional knockout mice (*CX3CR1-cre;K3$^{f/f}$/TGFβ1$^{+/+}$*). Deep vascular plexus of *CX3CR1-cre;K3$^{f/f}$/TGFβ1$^{f/f}$* and *CX3CR1-cre;K3$^{f/f}$/TGFβ1$^{+/+}$* mice stained with isolectin and pSMAD3. **e**, **f** Quantifications of pSMAD3 levels from deep vascular plexus (two-tailed *t*-test; $P < 0.0001$ (***); $N = 5$ mice) and vascular loops (one-way ANOVA with Tukey's post hoc; $P < 0.0001$ (***), $P = 0.0473$ (*); $N = 7$ mice). **g** P16 retinas stained with isolectin as superimposed and a cross-sectional view of the vascular layers is shown. The white arrowheads indicate neovascular sprouts invading the ONL. **h** The density of excessive neovascular sprouts is shown (one-way ANOVA with Tukey's post hoc; $P < 0.0001$ (***), $P = 0.7573$ (ns); $N = 9$ mice). All bar graphs are represented as mean and error bars are SEM.

**Microglial culture on hydrogels**. The hydrogel was prepared using the Hystem Cell Culture Scaffold Kit (Sigma) according to the manufacturer's protocol. Briefly, the Hystem stock solution was mixed with Extralink stock solution (crosslinker) along with 10 μg/ml concentration of fibronectin (Corning). The mixture was then poured into a 12-well plate and incubated at 37 °C for gelation to occur. The rigidity of the hydrogel was modified by changing the concentration of the crosslinker to obtain approximate gel stiffnesses of 60 and 600 Pa. The gels were pre-incubated with media and microglia were plated on top of the gel and allowed to spread overnight.

Silicone gels in 6-well plates were commercially purchased from Softsubstrates. com. The gels were coated with 10 μg/ml of fibronectin (Corning). The gels were pre-incubated with media and microglia were plated on top of the gel and allowed to spread overnight.

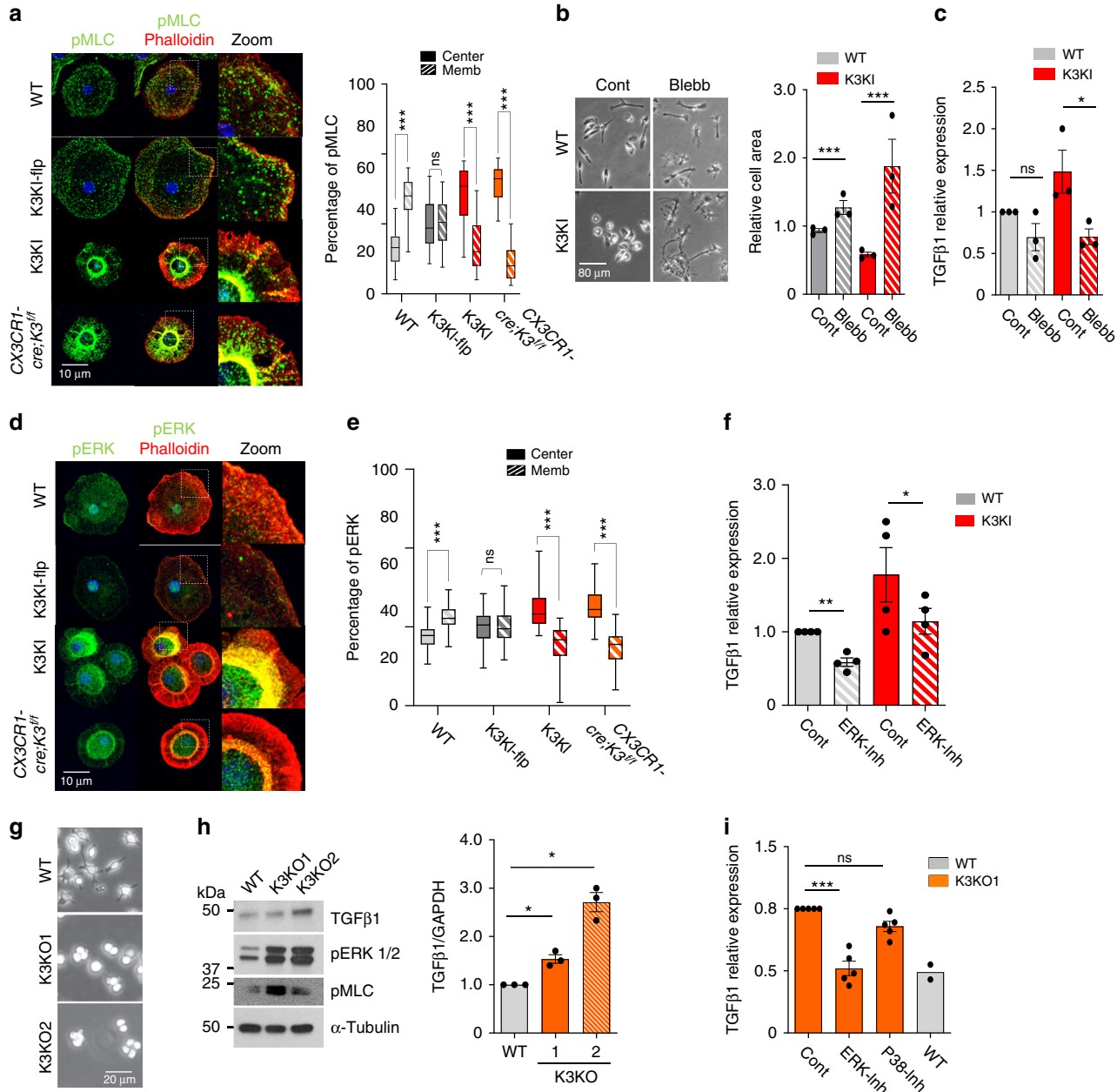

**Fig. 7 Kindlin3 deficiency leads to mislocalization of myosin light chain II and increased ERK activation. a** Immunostaining of primary microglial cells for phosho-myosin light chain II (pMLC) and Actin (phalloidin). The z-planes of confocal microscopy images were stacked to produce a superimposed image. An enlarged view of the cell edge shows co-localization of pMLC and actin to the membrane. The bar graph shows the proportion of pMLC localized either to the membrane (outer one-third area) or center (inner two-third area) of the cell (two-tailed *t*-test; *P* < 0.0001 (***), *P* = 0.7780 (ns); *N* = 35 cells from three experiments). **b** Phase-contrast image of microglia treated with DMSO (control) or 50 µM blebbistatin (Blebb) and spread on fibronectin-coated plates for 12 h. Bar graph shows cell spread area relative to WT control (two-tailed *t*-test; *P* < 0.0001 (***); *N* = 3 experiments). **c** QPCR analysis for TGFβ1 in control vs. blebbistatin-treated microglia (two-tailed *t*-test; *P* = 0.0684 (ns); *P* = 0.0234 (*), *N* = 3 experiments). **d** Immunostaining of microglia for phospho-ERK1/2 (pERK) (green) counterstained with phalloidin for actin (red) and DAPI for nuclei (blue). **e** The bar graphs show the proportion of pERK localized to membrane or the cell center (two-tailed *t*-test; *P* < 0.0001 (***), *P* = 0.7535 (ns); *N* = 40 cells from three experiments). **f** TGFβ1 mRNA levels assessed by QPCR in WT and K3KI microglia treated with vehicle (Control) or ERK inhibitor (U0126) (one-tailed *t*-test; *P* = 0.0028 (**), *P* = 0.0307 (*); *N* = 4 experiments). **g** Representative phase-contrast images of WT and the two clones of Kindlin3 knockout RAW cells, K3KO1 and K3KO2, spread on fibronectin-coated plates from five independent experiments. **h** Western blot analysis of WT and K3KO RAW cell lysates for TGFβ1, p-ERK1/2, and pMLC. Equal protein loading is shown by α-tubulin. The bar graph shows TGFβ1 levels (one-way ANOVA with Dunnett's post hoc; $P_{WT-K3KO1}$ = 0.0406 (*), $P_{WT-K3KO2}$ = 0.0205 (*); *N* = 3 experiments). **i** TGFβ1 mRNA levels as assessed by QPCR in K3KO cells treated with vehicle, (U0126) and P38 inhibitor (Sb20358) (one-way ANOVA with Dunnett's post hoc; *P* < 0.0001 (***), *P* = 0.0543 (ns); *N* = 5 experiments). WT levels (*N* = 2) are shown only for an estimated comparison. All bar graphs are represented as mean and error bars are SEM.

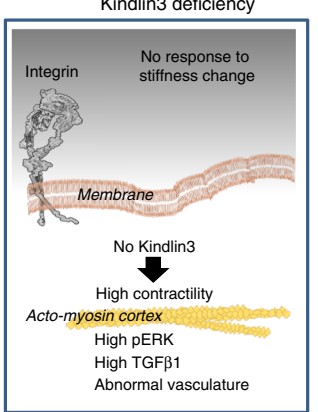

**Fig. 8 Schematic summarizing the main findings.** Kindlin3 is an intracellular integrin-binding adaptor molecule containing 3 FERM (F1, F2, and F3) domains and a PH domain. Kindlin3 binds integrins via its F3 subdomain and to the plasma membrane phospholipid, PIP2, through its PH domain. The FERM domains also interact with numerous cytoskeletal proteins and adaptors. ERK localizes to the actin cytoskeleton and is phosphorylated exclusively on actomyosin bundles in a myosin-II-dependent manner. Actomyosin bundles serve as a scaffold for activation of ERK signaling. In the absence of Kindlin3, the cytoskeletal network is dissociated from the membrane, resulting in contraction of myosin in the center of the cell. High myosin contractility leads to ERK phosphorylation, eventually resulting in overexpression of TGFβ1.

**RNA isolation and RT-QPCR.** Total RNA from cells was isolated using the Exiqon mirQury RNA isolation kit following the manufacturer's protocol. RNA concentration and purity were measured using a Nanodrop spectrophotometer. The RNA was diluted to a concentration of 200 ng/μl with RNase-free water and cDNA was prepared using a Qiagen RT-PCR kit. Briefly, the genomic DNA was eliminated by incubating RNA with RNase-free DNase for 2 min at 42 °C. It was then mixed with reverse transcriptase and random hexamers, and incubated at 37 °C for 1 h. The cDNA was diluted and used for PCR analysis. Quantitative real-time PCR was performed with the SYBR green master mix. The primers were used at a final concentration of 50 pM and the sequences are provided in the Supplementary methods table 1. Experiments were performed using an Applied Biosystems Quant studio 3, and data processing was performed using ABI SDS v2.1 software. The relative quantitation of each target mRNA was assessed using the comparative Ct method. GAPDH was used as an internal control.

**Mouse transcriptome array.** RNA was isolated from freshly enucleated retinas of control and *CX3CR1-cre;K3^{f/f}* mice of age P14. The RNA quality control followed by transcriptome array procedure and data analysis was performed by GEGF facility at the Case Western Reserve University. Briefly, 150 ng of total RNA at a concentration of 50 ng/μl was used to process the Affymetrix GeneChip® Mouse Transcriptome Assay 1.0 Arrays. Biotinylated cDNA was prepared using the GeneChip® WT Plus Reagent Kit according to manufacturer's instructions. Following fragmentation and labeling, 5.5 μg of cDNA were hybridized for 16 h at 45 °C on MTA Array. GeneChips were washed and stained in the Affymetrix Fluidics Station 450. GeneChips were scanned using Affymetrix GeneChip Scanner 3000. Affymetrix's software was used to normalize arrays via the Robust Multi-Array algorithm (RMA). The fold-change and *p*-values were calculated by ANOVA/empirical Bayes algorithms in the Affymetrix's Transcriptome Analysis Software. The list of genes with significant changes (with $p < 0.05$) were further analyzed by ConsensusPathDB at http://cpdb.molgen.mpg.de/ according to the protocol by Herwig et al.[62] for over-representation analysis of gene sets. From the resulting analysis, the top 3 overrepresented pathways (based on *q*-values calculated by ConsensusPathDB) are shown in the graph.

**Western blot analysis.** Cell protein extracts were lysed in RIPA buffer supplemented with protease inhibitor cocktail. The extracts were centrifuged at $16000 \times g$ for 20 min at 4 °C and the supernatant was heated with Laemmli buffer at 95 °C for 7 min. The proteins were separated on 12% polyacrylamide slab gels in a Mini-Protean II system. Proteins were then electrophoretically transferred to Immobilon-P, PVDF membrane (Millipore Corp., Bedford, MA) and blocked with 5% nonfat dry milk in TTBS (TTBS; 0.2 M Tris [pH 7.4], 1.5 M NaCl, 0.1% thimerosol and 0.5% Tween 20). The blots were washed with TTBS and incubated with primary antibody at 4 °C overnight. The following antibodies were used: Anti-TGFB1, pMLC, pERK1/2, pSMAD3, and SMAD3 at 1:1000 dilution. GAPDH and β-actin

were used at 1:3000 dilution. Blots were washed in TTBS and incubated with the appropriate secondary antibody conjugated to horseradish peroxidase at 1:5000 dilution for 1 h at room temperature. They were then washed with TTBS twice for 15 min each and four times 5 min each. The blots were developed with Signal fire chemiluminescence kit. Further antibody and reagents details are provided in Supplementary methods Table 1.

**Immunohistochemistry.** For retinal whole mounts, eyes were isolated into 4% PFA and kept on ice for 1 h. Whole retinal cups were isolated and fixed with 4% PFA overnight. The retinas were then washed and permeabilized with 0.5% Triton in blocking solution (5% goat serum, 3% BSA in PBS) overnight at 4 °C. Next, the retinas were washed in PBS and stained with primary antibodies at a 1:100 concentration in blocking solution overnight at 4 °C on a shaker. Following washes with PBS, they were incubated with Alexa fluor-conjugated secondary antibodies in blocking solution overnight. Alexa fluor 568-conjugated isolectin was used at 1:200 to visualize blood vessels. After final washes in PBS for 3 h, retinas were flattened and mounted on glass slides with Prolong Gold Antifade Mountant.

For retinal cross sections, whole eyes were fixed in 4% PFA overnight at 4 °C, washed in PBS, and transferred to 30% sucrose at 4 °C overnight. The fixed eyes were transferred to a cryomold and embedded in OCT. Sections (10 μM) were prepared from the molds with a cryotome. The sections were permeabilized with 0.5% triton, 5% goat serum, and 3% BSA in PBS for 1 h, washed with PBS, and then incubated in a blocking solution (5% goat serum, 5% BSA in PBS) for 1 h at room temperature in a humidified chamber. Next, the sections were incubated with primary antibodies at 1:200 dilutions (in blocking solution) at 4 °C overnight in a humidified chamber, followed by three 5 min-washes in PBS. Sections were then incubated with Alexa Fluor-conjugated goat anti-rabbit secondary antibody for 1 h at room temperature in a dark humidified chamber. Negative control sections were incubated with secondary antibody alone. Finally, the sections were washed three times for 5 min in PBS, counterstained with Hoechst or DAPI in PBS for 10 min, followed by a final wash in PBS. Sections were then mounted with Prolong Gold Antifade Mountant and examined using a fluorescence or confocal microscope.

**Immunocytochemistry.** Purified microglia were plated into a six-well plate with glass coverslips at the bottom. The plate was centrifuged at $700 \times g$ for 3 min at room temperature to force cells to adhere to the coverslip simultaneously. Immediately, the cells were fixed with 4% PFA for 20 min to prevent further cell spreading. This was done to keep the cell size/foot-print equal between WT and Kindlin3-deficient microglia for comparison of immunostaining between genotypes. In contrast, because WT cells spread more than Kindlin3-deficient cells, comparison of immunostaining intensity could be imprecise. The PFA was washed away with PBS and the cells were permeabilized with 0.5% Triton in blocking solution for 10 min, and incubated in blocking solution for 1 h at room temperature. The following antibodies were used at a concentration of 1:100 in blocking solution overnight at 4 °C in a humidified chamber: anti-phospho S19 MLC2 and pERK1/2. Alexa-conjugated phalloidin at 1:200 dilution was used to visualize actin. Fluorescently-conjugated goat anti-rat and mouse secondary antibodies were used at 1:500 dilution. Finally, the cells were washed and mounted onto slides with Vectashield containing DAPI. Further antibody and reagents details are provided in Supplementary methods Table 1.

**Chondroitin sulfate treatment of retinas.** Mice were euthanized and eyes were immediately isolated into ice-cold PBS followed by retina isolation. The freshly isolated retinas were transferred to DMEM/F12 (with 20% FBS, 100 U/ml penicillin and streptomycin, 0.25 μg/ml amphotericin B and supplementation of non-essential amino acids (NEAA)) with or without 20 mg/ml CS (Selleck Chemicals LLC). Retinas were incubated at 37 °C in 5% $CO_2$ for up to 6 h. The retinas were then immediately analyzed by AFM. Confocal imaging of microglia in these retina was either performed immediately after treatment or after fixation in 4% PFA overnight.

**Microscopy and image analysis.** Confocal images were obtained using a Leica SP5 confocal/multi-photon microscope in the Imaging Core of the Lerner Research Institute. Retinal whole mounts were imaged with ×25 and ×40 objectives and a step size of 2–3 μm for vasculature and 0.5–1.0 μm for in vivo microglia. In vitro cell were imaged with ×63 objective and a step size of 0.5–1.0 μm. The images were processed and staining intensity was quantified using Volocity software and Image J. The 3-D images were prepared by merging the confocal stacks using Volocity. The retinal loop area, junctions/field, and microglia numbers were quantitated using FIJI Image J software. The distances of pMLC and phalloidin staining from the cell membrane were analyzed using ImagePro software.

Cell polarity was measured using Image J. The following steps were involved for in vivo analysis: (i) image processing to separate microglial cells from background; (ii) construction of a skeleton to represent the spatial structure of cell bodies and branched processes if required; (iii) generation of dendritic tree area to identify the longest and widest axis; (iv) measurement of microglial length in pixels, including cell body and the branches that are along the longest dendritic tree axis. Similarly, the width was measured along the widest axis: (v) division of length by width to get the ratio, i.e., polarity.

**Atomic force microscopy**. All measurements were made using an MFP-3D-Bio AFM mounted on an inverted optical microscope. For characterization of fixed and live retinal tissues, a 4.5 or 35 μm bead, respectively, was glued to an Arrow$^{TM}$ TL1 (spring constant ~0.03 N/m) tipless cantilever. The probe was submerged in PBS prior to each experiment to stabilize, minimize drift, and achieve thermal/mechanical equilibrium. Retinal sections (40-μM thick) were prepared from the molds with a cryotome. Force-indentation curves ($n \geq 15$ in each region for fixed retinal tissues, and $n \geq 55$ for unfixed retinal tissues) were obtained with a trigger force of 4 nN at an approach speed of 5 μm/s. Step sizes of 20 μm for retinal sections immersed in PBS and 50 μm for live tissues immersed in media were used. The force-indentation curves were fitted to a Hertz model to calculate the elastic modulus using Igor Pro 6.37 software.

**ELISA**. Microglia were isolated as described above and plated into 6-well plates at equal numbers. They were serum-starved for 24 h, fresh FBS and phenol free media was added, and the conditioned media was collected at 24 and 48 h time points. The conditioned media was centrifuged at $10,000 \times g$ for 20 min at 4 °C to remove any cell debris and then diluted at 1:500. Microglia on the plates after media collection were lysed with RIPA buffer with ROCHE protease inhibitors and centrifuged at $16,000 \times g$ for 20 min at 4 °C. The conditioned media and cell lysates were analyzed for TGFβ1 levels using the TGFβ1 Emax immono assay kit according to the manufacturer's protocol.

**CRISPR-Cas9-mediated knockout of Kindlin3**. CRISPR-Cas9 technology was used to generate Kindlin3 knockout in a Raw 264.7 cell line. Two independent sgRNAs were designed by the CRISPR Design Tool[63]. Annealed oligonucleotides were ligated into the vector LentiCRISPRv2 vector (Addgene) digested with BsmBI (Fermentas). Pheonix Packaging cells (Takara) were transfected with LentiCRISPRv2 sgRNAs by lipofectamine3000 (Thermofisher) for 48 h. Then, the Raw 264.7 cells were infected by lentivirus from the culture medium with Lenti-X™ Packaging Single Shots (Clontech) according to the manufacturer's protocol. Seventy-two hours later, cell selection was performed in the presence of 2 μg/mL puromycin (Thermofisher) for 48 h. The puromycin-resistant cells were collected and sorted by a flow cytometer into 96-well plates. Individual clones were examined by western blotting or genomic DNA sequencing. sgRNA hairpin sequences are provided in the Supplementary methods Table 1.

**Study approval**. Animal experimental procedures were performed in accordance with National Institutes of Health (NIH) guidelines on animal care and all protocols were approved by the Institutional Animal Care and Use Committee at the Cleveland Clinic. The lentivirus infection was performed in accordance with a Cleveland Clinic IBC protocol.

**Reporting summary**. Further information on research design is available in the Nature Research Reporting Summary linked to this article.

## Data availability
All data generated or analyzed during this study are included in this published article (and its supplementary information files). The source data for Figs. 1d, f–i, 2b–d, f, h–i, k–m, o–p, 3d–e, g, 4b, d, f–g, i, k, 5b–e, i, 6b–c, e–f, h, 7a–c, e–f, h–i and Supplementary Figs. 1b, f, g, 2c, e, 3a-c, 4b, e, 5a, c–g, 6a, b, 7a-g, i, 8a, d, b, 9e, g, 10a-c, e–f are provided as a Source Data File.

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

## Acknowledgements

This work was supported by grants from NIH; R01 HL071625 to T.V.B. and R01 HL077213 to E.P. We acknowledge the use of the Cleveland Clinic Imaging Core equipment and services supported by NIH SIG grant 1S10RR026820-01. We would also like to thank Chris Nelson for proof-reading the paper and Chase Bertagnolli for assistance with data analysis.

## Author contributions

T.D. designed and performed experiments, wrote the paper, J.M., G.M., I.Z., H.L., S.S., and C.W. performed experiments. E.P. and C.K. performed data analysis, wrote the paper. T.V.B. conceived experiments, analyzed data, wrote the paper, and secured funding.

## Competing interests

The authors declare no competing interests.
