## [Peer Review File · Nature Communications]

Reviewers' comments:

Reviewer #1 (Remarks to the Author):

The authors report that microglia use Kindlin3 to sense increasing tissue stiffness in the developing retina, resulting in downregulation of TGF β 1 expression and subsequent inhibition of vascular sprouting into ONL. Upregulation of TGF β 1 and excessive sprouting of the retinal vessels was observed in gene targeted mice where Kindlin3 was deleted postnatally in microglia, as well as in constitutive, ubiquitous Kindlin3 knock-in mice expressing low levels of mutant Kindlin3 that is unable to bind integrins. The vascular pathologies could be corrected by either microglia depletion or by microglia-specific knockout of TGF β 1 in the mutant mice. In vitro, Kindlin3 depletion impaired polarization of microglia on stiff substrates, correlating with altered localization of P-MLC, P-Erk and filamentous actin and increased TGF β 1 expression. In summary, this is a well-performed study supporting the novel conclusion that Kindlin3 functions as a mechanosensor in microglia to downregulate paracrine TGF β 1 signaling, thereby limiting vascular growth.

Specific comments:

1. It is suggested that microglia signal in a paracrine manner to endothelial cells to limit sprouting at the stiffness boundaries (e.g. p. 13, row 280, p. 14, row 317). However, the vascular system appears to develop normally in the absence of microglia and microglia-derived TGF β 1 (Fig.6a, c, d, g, f). Thus, is it only that the absence of Kindlin3 in microglia induces paracrine signaling that leads to aberrant vascularization, whereas in the WT retina, microglia are not needed for limiting vascular growth into ONL? Please clarify.
2. It is concluded that Kindlin3 orchestrates stiffness sensing of microglia in an integrin-independent manner, however, the mechanisms remain open. In fact, the density of the retinal vasculature in CX3CR1-cre; β 1f/f mice (Fig. 5h) seems comparable to the CX3CR1-cre;K3f/f mice (Fig. 5a). Vascular loops, in addition to sprouts should be quantified. Microglial polarization in CX3CR1-cre; β 1f/f mice was also slightly decreased compared to WT – were the differences statistically significant (Fig. 2g, indicate significances also in 2k)? Were integrins involved in in vitro adhesion/polarization of microglia?
3. It has been previously reported that microglia were absent in the CNS of TGF β 1-deficient mice (Butovsky, Nat Neuroscience, 2014). Quantification of microglia in CX3CR1-cre;K3f/f/TGF β 1f/f and CX3CR1-cre;TGF β 1f/f mice would confirm that the rescue was not mediated via decreased microglia numbers.
4. It was suggested that TGF β 1 might exhibit an autocrine effect on microglia, but this was not investigated. Since several previous publications (e.g. Ma et al. eLife 2019) have reported various effects of TGF β signaling in microglia, the phenotype of microglia in CX3CR1-cre;K3f/f/TGF β 1f/f mice is of interest. Is microglia polarization in vitro affected by TGF β 1?
5. As reported earlier (Xu et al. ATVB, 2014), Kindlin3 protein levels in K3KI mutant mice were equal to those in WT mice, whereas in the current study the authors report decreased mRNA levels in

microglia of K3KI mice compared to WT or K3KI-Flp mice (Fig. 2h). It should be clarified if the nomenclature is the same between the two studies. In addition, Kindlin3 protein levels should be analyzed in the mutant mice.

6. It has been previously reported that the K3KI mice have significantly increased numbers of T cells and slightly increased numbers of red blood cells (Xu et al., *ATVB*, 2014). Considering that in the current study the authors used K3KI mice with significantly reduced levels of Kindlin3 mRNA, the authors should discuss the potential involvement of other cells types.

7. It is not clear if the vascular phenotype is transient or persistent beyond P16 in CX3CR1-cre;K3f/f and K3KI mice.

8. The difference in tissue stiffness between INL and ONL was 3-fold, whereas a 10-fold difference in substrate stiffness was used to polarize microglia in vitro (Fig. 2a,b). In addition, the stiffness experienced by microglia was much higher in vivo than in vitro. The authors should discuss these differences in the experimental conditions between in vivo and in vitro.

9. Fig. 3h. The secreted TGF β 1 levels are measured by ELISA after 24h and 48h of culture. Why the TGF β 1 levels are not increasing over time in WT cultures?

10. Fig. 4c. Quantification of the pSMAD3+ lectin+ cells in K3KI and WT mice would strengthen the conclusions.

11. Fig. 7a, d. How many cells per experiment were quantified, when the localization of pMLC and pERK were analyzed? Are the differences statistically significant?

Minor comments:

1. Fig. 3e-g. The labels should be standardized. In Fig. 3e the gene of interest is on a y-axis, in Fig. 3f, g the gene of interest is above the diagram.

2. Fig. 6b-c. The color coding of the bars should be included in the figure.

3. Fig. 6f. The loops should be quantified per 500 μ m² as in previous figures for easier comparison.

4. A typo in row 519 should be corrected, polaization to polarization

5. A typo in row 808 should be corrected, VEGEFA to VEGFA

6. The authors might consider some of the earlier studies where the effects of TGF β 1 on microglia quiescence and ramification were investigated (Abutbul, *Glia*, 2012, Schilling, *Eur J Neuroscience*, 2001)

Reviewer #2 (Remarks to the Author):

The manuscript by Dudiki and colleagues investigate the effect of tissue stiffness in the cellular and molecular responses of microglia and their consequences to the process of neovascularization. The authors found that microglia respond to tissue stiffness in the retina and that this requires kindlin3. Furthermore, they demonstrate that deletion in K3 or inactivation of the K3KI domain results in increased TGF- β secretion and pSMAD in the endothelium. At the tissue level, the retinal vascular responds by increasing sprouting and promoting hypergrowth. Overall the study brings to light a previously unexpected role of kindlins, namely in their ability to regulate TGF- β with extremely interesting consequences to the cellular behavior of microglia and endothelial cells. The authors provide clean and stunning images, furthermore the rescue experiments offer further support to the main conclusions and bring the story to a very nice closure. However, there are several aspects that deserve further evaluation, in particular, some controls are missing, editorial (or experimental) changes should be made to ensure that the stated conclusions remain true to the findings,

1. The stated conclusion that: “the microglia mechanosensing is independent of integrins” has not been fully supported by the data. What the authors show is that: (a) Kindlin3’s effect on integrins is not required for mechanosensing, instead the needed region is the KI domain of this intracellular adaptor protein and that (b) beta 1 and 2 integrins are not required (although controls for these KOs are needed – see below) but the authors did not evaluate all integrins. In order to state that integrins are not part of this mechanism, additional experiments are needed, as well as a comprehensive evaluation of each integrin expressed by microglia, which expand beyond the beta 1 and beta 2 families. Alternatively, this reviewer would suggest to revise the statement to bring more specificity and alignment with what has been shown, instead of making a broad generalization that does not reflect the experiments included in the study.

2. Along the lines with #1, the statement “microglia respond to tissue stiffness in integrin-independent fashion” needs editorial clarity and needs to be better controlled. The authors must provide evidence that in fact all the KOs shown in panel “f” had absence or significant reduction of the protein being targeted by Cre at the genomic level.

3. The authors have convincingly shown that TGF- β levels change by two to three-fold in the K3KI microglia. While this is impressive, it is unclear to what extent this impacts signaling. For example, in vitro, addition of 1ng/ml or 10ng/ml of TGF- β have a similar outcome in activation of Smad3 and 4. This is because the rate-limiting constituent are the receptors, rather than the ligand in this signaling cascade in vitro (this is not the case for VEGF, however). Can the authors assess whether this two-three fold change in TGF- β is sufficient to translate into a Smad3/4 signaling? This reviewer appreciates the results shown in Figure 4a, however, this does not directly prove that the differences in pSMAD3 relate to TGF- β , instead, it is possible that levels of SMAD3 or TGF- β receptors might be distinct in K3KI microglia (a comprehensive transcriptional profile would be important – as also stated below).

4. While the selection of TGF- β was logical and the authors have also shown TNF α , VEGF, sflt 1 and IL1 β , all of these have been hand-picked. Considering the availability of technology today, it will be important to comprehensively profile WT and Kindlin3-KO microglia and determine if indeed

these are the key factors that affect EC biology. Such unbiased approach is important to offer support to the conclusions put forward.

5. Controls for Figure 6 – while the levels of pSMAD3 are impressive in the image provided, imaging for assessment of levels is not the most reliable approach. Instead, Western blots showing total SMAD3/ pSMAD3, will unequivocally support the statements put forward and contribute to address some of the previously raised concerns. As discussed previously in relation to integrins, the authors must provide assessment of the level of the Cre deletion on protein (for TGF- β in the case of figure 6).

6. The experiments presented in Figure 7 contribute to address a key question in the manuscript, namely: how could kindlin3 regulate TGF- β 1's secretion in microglia. Using blebbistatin, the authors normalized the hypercontraction of microglia and the levels of TGF- β . Were the transcriptional levels of TGF- β also changed/normalized? (Figure 3e shows a 3.5 fold increase in K3KI).

Also in this figure, it is puzzling that the levels of secreted TGF- β in Figure 3h are 2 to 3 fold increased in K3KI, while in Figure 7 this does not reach 1.5 fold (average). Can the author comment on this discrepancy?

7. In several occasions, the authors mention retinal angiomatous proliferation (RAP), the impact of the present manuscript would significantly benefit from comparing the vasculature pattern of K3KI mice to RAP specimens. Also, Are the levels of TGF- β comparable? or What is the degree of pSMAD3 in the endothelium?

Minor Concerns:

- In the summary, the opening statement "Tissue mechanics regulate development and virtually all pathologies of the Central Nervous System (CNS) by largely unknown mechanisms" is a bit too bold. While it is clear physical forces are extremely important in biological tissues, the role of tissue mechanics in development and pathology of the CNS has not been sufficiently investigated to make such statement. Some might even say that Alzheimer's Disease for example, a rather prevalent pathology of the CNS does not require any contribution by tissue mechanics for the onset of the disease – despite the fact that at later stages the presence of plaque, as stated by authors, thickness of retina might later (as a consequence of the disease) alter physical stiffness of the neural tissue. In any case, there is no need to start with a statement that might turn investigators away from reading the rest of the paper. I strongly suggest re-writing.

- In Figure 4c, the use of colors (red and magenta) are unfortunate. To better see colocalization, pls. do change red to green or blue. This should be trivial with the original images.

Reviewer #3 (Remarks to the Author):

The manuscript by Dudiki et al. aimed to investigate microglial mechanosensing and its role in regulating vascular patterning in the retina. The authors conclude that microglia migrate through the tissue stiffness gradient, change their morphology, and subsequently regulate vascularization through a Kindlin3 and subsequent TGF β 1 signaling mechanism. The idea that microglial morphology and function is regulated by a mechanosensing mechanism is interesting and novel and elucidating microglial functions in vascular development is also important. However, there are major concerns with this paper regarding whether the data support the major conclusions and the connection between microglial mechanosensing and vascular regulation is weak. These and other concerns are outlined below:

- 1.) The conclusion that microglia change morphology based on tissue stiffness in vivo is not convincing. The authors show correlative changes in microglial morphology within tissue with different stiffnesses. They also showed in vitro that modulating the substrate stiffness can affect microglial morphology. However, all these data are correlative and do not provide sufficient evidence in vivo that tissue stiffness directly regulates microglial morphology. In vitro systems are quite different than in vivo settings and are not sufficient to support the author's claim in vivo. The authors would need to modify tissue stiffness in vivo and show that it directly regulates microglial morphology through microglia-specific receptors.
- 2.) The authors propose that microglial mechanosensing occurs through Kindlin 3, but they provide no evidence that this is a mechanosensing receptor. More work is required to show Kindlin 3 is a mechanosensing cell adhesion molecule and it would be important to determine the extracellular ligand. It is unclear what is binding Kindlin3.
- 3.) The Kindlin 3 mutant mice potentially have a very interesting microglial phenotype, but this does not seem to be specific to the cell polarity they measure or mechanosensing function. In addition to changes in cell polarity, the microglial cells in these mutant mice appear to have significant changes in their branch numbers, soma size, and cell numbers. The authors state that cell numbers were no different in the mutant mice, but this does not seem to be the case in the images they provide and they provide no quantification of these data. This requires more assessment of microglial phenotypes and the role for Kindlin 3 in modulating these phenotypes, which are less likely due to changes in mechanosensing. It would also be helpful to assess microglia morphology in these mutants more globally in the retina and in the CNS.
- 4.) The authors cannot conclude that their effects are microglia specific. Perivascular macrophages, which have a similar lineage to microglia and also express CX3CR1creER, likely also express Kindlin3 and TGF β 1. Therefore, their effects may not be microglia-specific. To better support their conclusions, the authors should provide evidence (e.g. in situ and/or immunostaining) that Kindlin 3 is restricted to only microglia in the retina.
- 5.) There is literature that demonstrates a role for TGF β 1 signaling in regulation of the vascular (Shih et al. 2003, Walshe et al. 2009, Arnold et al. 2014). Additionally, there is also published literature showing that microglial depletion causes changes to vascular complexity (Fantin et al. 2010). Therefore, without further assessment of the Kindlin3 mechanism, the study lacks novelty.

6.) The signaling between Kindlin3 and TGF β 1 to regulate the vasculature requires more work. All mechanistic work assessing downstream signaling from Kindlin 3 was performed in vitro. It is unclear if this translates in vivo. Further, the authors show inconsistent data related to pMLC in vitro (Fig 7h) where K3KO1 shows increased pMLC, but this is not replicated in K3KO2. The levels of pMLC appear to be negatively correlated with TGF β 1 levels in the K3KOs.

7.) The developmental time course of microglial morphology changes in the deep versus intermediate vascular layers requires clarification. The authors suggest that microglia morphology is regulated by tissue stiffness and that the cells on the deep layers are on the stiffest ONL. However, these cells also are in contact with the less stiff IPL and they are located within the intermediate vascular layer, which has an even lower degree of stiffness than the IPL or ONL. Also, does the stiffness of the layers change across development? If not, it is unclear why microglia would have similar cell polarity in the intermediate (less stiff) and deep (more stiff) vascular layers early in development.

8.) The authors conclude that microglia are directionally migrating, but they show no time-lapse imaging to show migration. This terminology should be modified.

9.) The methods described for microscopy require more detail. For example, how were images acquired for a given experiment, what magnification was used, were z-stacks collected, how large were the z-steps etc.

10.) While n's are given in the figure legend, it is frequently difficult to determine if the authors are referring to fields of view, animals, cells, etc. There is also large spreads in the n's. For example, in some experiments the n=5-7(Fig 1d) where as in other experiments the n=36 (Fig 1f), 60 (Fig 2b) etc. As it is written, the authors say this is per group. Does this mean 36, 60, etc. mice were quantified? If this reflects cell number, this is not appropriate and statistics should be re-run with n's defined as animals. If this is animal number, it is unclear why so many mice were used for one experiment and not others.

11.) The statistics run in this manuscript appear to be incorrect. The authors state that used an unpaired t-test as their posthoc analyses following ANOVAs. The statistics should be re-run and a posthoc analysis such as Tukey's or Bonferroni's should be used.

12.) There are many grammatical errors throughout, which makes the manuscript a difficult read. This requires significant editing.

Responses to reviewer comments

We would like to thank the reviewers for their careful analysis and consideration of our work. To address their comments, we had performed a series of new experiments, adding new results into the main figures whenever possible and into the Supplementary Figures otherwise. Collectively, we have added nearly 40 panels of new data. Of particular importance, we now show and characterize the main vascular abnormalities at 21 and 60 days. Remarkably, the phenotype becomes even more dramatic at 21 days, as there are numerous deep vascular malformations reaching below the layer of photoreceptors and nearly forming a “fourth” vascular layer. This severe pathology is likely to be further exacerbated in Kindlin3-deficient patients prone to severe bleeding. Among other experiments, we had optimized a new technique developed by Franze’s group for neural tissues to alter tissue stiffness in retinas. We show that softening of retinal layers (as verified by atomic force microscopy) diminished polarization of wild-type (WT) microglia, whereas Kindlin3-deficient microglia were not responsive. We have also performed *ex vivo* experiments with microglia in silicone gels (in addition to HA hydrogels), where stiffness closely mimics retinal layers. None of these techniques were previously applied to microglia or retina and are technically challenging and time-consuming (primary microglia are isolated from P1 brains with a low yield). Besides three Kindlin lines (Microglia-specific knockout and Integrin-incompetent Kindlin QW Knock-in mutants with 100% and low expression), and a rescue mouse model, i.e. microglia-specific inducible double KO of Kindlin3 and TGF β , we analyzed two integrin KO lines; β 2 and β 1 (the latter being specifically made for this study) to further support our conclusions. We have included a brief summary of RNAseq and gene array analyses, which together with ELISA-based assays aided in the identification of microglial TGF β as a culprit for our vascular phenotype. The details of these new experiments and point-by-point responses are presented below.

Reviewer #1:

The authors report that microglia use Kindlin3 to sense increasing tissue stiffness in the developing retina, resulting in downregulation of TGF β 1 expression and subsequent inhibition of vascular sprouting into ONL. Upregulation of TGF β 1 and excessive sprouting of the retinal vessels was observed in gene targeted mice where Kindlin3 was deleted postnatally in microglia, as well as in constitutive, ubiquitous Kindlin3 knock-in mice expressing low levels of mutant Kindlin3 that is unable to bind integrins. The vascular pathologies could be corrected by either microglia depletion or by microglia-specific knockout of TGF β 1 in the mutant mice. In vitro, Kindlin3 depletion impaired polarization of microglia on stiff substrates, correlating with altered localization of P-MLC, P-Erk and filamentous actin and increased TGF β 1 expression. In summary, this is a well-performed study supporting the novel conclusion that Kindlin3 functions as a mechanosensor in microglia to downregulate paracrine TGF β 1 signaling, thereby limiting vascular growth.

We thank the reviewer for these constructive comments and questions, which are addressed in detail below.

1. It is suggested that microglia signal in a paracrine manner to endothelial cells to limit sprouting at the stiffness boundaries (e.g. p. 13, row 280, p. 14, row 317). However, the vascular system appears to develop normally in the absence of microglia and microglia-derived TGF β 1 (Fig.6a, c, d, g, f). Thus, is it only that the absence of Kindlin3 in microglia induces paracrine signaling that leads to aberrant vascularization, whereas in the WT retina, microglia are not needed for limiting vascular growth into ONL β Please clarify.

While microglial ablation experiments show that microglia are important for vascularization, their exact role might depend upon the stage of retinal development. We have now added new results showing that depletion of microglia at early stages delays vascularization in WT mice and, therefore, is clearly important for retinal vasculature (see Fig.S7c and figures summarized for the reviewers R1:1). After P14, however, the lack of microglia in WT no longer diminishes vasculature, instead resulting in a modest increase in vascular density (Fig.S7d and R1:1). Thus, in agreement with previous reports, microglia in WT retinas have both angiogenic (at the earlier stages before P12-14) and anti-angiogenic (at later stages) functions. We have also included a citation showing that TGF β expression during vascularization in retinas follows a bell-shaped response, gradually increasing and reaching a peak at P7-P8 and then plateauing and decreasing at P12 (just preceding the completion of the three layers).

In our experiments, the timing of pexidartinib treatment was very important, since its effect on microglia is not immediate. We treated mice with pexidartinib at P5-P12, followed by retinal harvesting at P14. Microglial depletion was gradual, resulting in >95% depletion 5-6 days after treatment initiation (P10-P11) and rescuing overgrowth in K3KO (Fig 6a). However, if the depletion of K3KO microglia occurred after P15-P16, then rescue was not achieved (data not shown).

Likewise, the timing of TGF β 1 deletion in Kindlin3 KO (overexpressing TGF β 1) is also critical. Note that *CX3CR1-cre;K3^{ff}/TGF β 1^{ff}* mice were fed with tamoxifen from P1-P5 to induce the simultaneous deletion of Kindlin3 and TGFB1 from microglia. As shown, this deletion prevented overgrowth of vasculature, but not vascular development, in K3KO mice. We believe that there are two reasons for this: a) Deletion of TGF β 1 is not immediate, and b) other factors might compensate for the lack of TGF β 1. Even global TGF β knockout shows differential effects on retinal vasculature, affecting vertical growth most dramatically². We have modified our text accordingly (lines 251-258).

R1:1

2. It is concluded that Kindlin3 orchestrates stiffness sensing of microglia in an integrin-independent manner, however, the mechanisms remain open. In fact, the density of the retinal vasculature in CX3CR1-cre; $\beta 1f/f$ mice (Fig. 5h) seems comparable to the CX3CR1-cre;K3f/f mice (Fig. 5a). Vascular loops, in addition to sprouts should be quantified. Microglial polarization in CX3CR1-cre; $\beta 1f/f$ mice was also slightly decreased compared to WT – were the differences statistically significant (Fig. 2g, indicate significances also in 2k)? Were integrins involved in *in vitro* adhesion/polarization of microglia?

The conclusion of integrin independence was made based upon Kindlin3 Knock-in mice bearing a mutation disrupting Kindlin3-integrin (including all integrins) interactions (K3KI-flp with normal Kindlin3 expression). Together, we analyzed 3 types of Kindlin3 mice: 1) K3KO in microglia, 2) Kindlin3 knock-in lacking integrin binding site (K3KI-flp), and 3) Kindlin3 knock-in mice with low Kindlin3 expression (K3KI). The overgrowth was only present in mice where Kindlin3 expression was altered, whereas in knock-ins expressing “integrin-incompetent” Kindlin3 (K3KI-flp), the vasculature was normal. Thus, based on these three mouse lines, we concluded that Kindlin3-integrin interactions are not required for microglia-mediated control of neovascular growth into ONL. These results are shown in Fig.5 and are discussed on page 11. Likewise, neither of the individual integrin $\beta 1$ or $\beta 2$ knockout mice exhibited vascular overgrowth into ONL.

The reviewer is correct that in contrast to the most abundant $\beta 2$ integrin, microglia-specific $\beta 1$ integrin KO has a mild vascular phenotype; however, the dramatic misdirection of vasculature observed in Kindlin3 KO is absent.

As per this suggestion, we have now included an in-depth characterization of $\beta 1$ integrin KO mice. While there was a significant increase in the number of loops, vascular growths into the ONL that are characteristic of TGF β overexpression were not present (Fig.5 h,i and Fig.S7a,b). Also, while there were morphological changes in microglia, their polarization was not as drastically affected as in the K3KI or K3KO (Fig.2f,g and R1:2d,e). *In vitro*, $\beta 1$ integrin KO microglia showed a small decrease in polarization (Fig.S1g). However, in contrast to K3KO, TGF $\beta 1$ expression in

$\beta 1$ KO microglia was not affected (Fig. R1:2g). Microglia from $\beta 2$ KO mice did not exhibit any changes in morphology either *in vivo* or *in vitro* (Fig.S1g and R1:2h).

3. It has been previously reported that microglia were absent in the CNS of TGF β 1-deficient mice (Butovsky, *Nat Neuroscience*, 2014). Quantification of microglia in CX3CR1-cre;K3f/f/TGF β 1f/f and CX3CR1-cre;TGF β 1f/f mice would confirm that the rescue was not mediated via decreased microglia numbers.

As suggested, in order to show that the rescue was not mediated by decreased microglial numbers, we have quantified microglial numbers in CX3CR1-cre;K3f/f/TGF β 1f/f and CX3CR1-cre;K31f/f mice. No significant differences were observed in microglial numbers as shown in Fig. S.7f,g and R1:3/4.

Similar to several other studies, Butovsky et al. (*Nature Neuroscience*, 2014) observed high expression of TGF β 1 and Tgfbr1 in microglia and lower levels in both neurons and other glial cells. Global knockout of TGF β 1 results in the absence of microglia from the CNS, leading to the conclusion that autocrine/paracrine signaling of TGF β 1 on microglia is critical for their development and maintenance of microglia-specific gene signature. The subsequent study from the same group (Lund et al., *Nature Immunology*, 2018) using tissue-specific and inducible TGFBR1 demonstrated that TGF β 1 signaling is not crucial for microglial survival, but is important for repopulation of the microglial niche by monocytes. Further, another study mentioned by the reviewer below by Ma et al. (*eLife*, 2019) shows that ablation of TGF β 1 signaling in microglia using TGFBR2 knockout in retinal microglia also did not result in a decrease in microglial numbers.

We used microglia-specific knockout of TGF β 1, which was induced postnatally, when CNS population by microglia has been completed. Moreover, TGF β 1 KO was created on top of K3KO, where TGF β 1 is overexpressed. As described above, both the timing and tissue specificity of this rescue experiment are important. As a result, microglial numbers were not affected. Together, the absence of microglia in the CNS observed in global KO could be attributed to either direct or indirect consequences of global TGF β 1 depletion on microglial migration to CNS in development. We have cited additional papers and modified the discussion.

4. It was suggested that TGFβ1 might exhibit an autocrine effect on microglia, but this was not investigated. Since several previous publications (e.g. Ma et al. eLife 2019) have reported various effects of TGFβ signaling in microglia, the phenotype of microglia in CX3CR1-cre;K3^{fl/fl}/TGFβ1^{fl/fl} mice is of interest. Is microglia polarization *in vitro* affected by TGFβ1?

The requested results are shown in (Fig.S7f,g) The CX3CR1-cre;K3^{fl/fl}/TGFβ1^{fl/fl} microglia appear similar to CX3CR1-cre;K3^{fl/fl} in terms of polarization, demonstrating that it is Kindlin3 that is responsible for polarization. The lack of TGFβ reversed the consequences of this polarization defect, but not the polarization itself. Overall, the *in vivo* morphology of microglia in a double KO mice is shown in Fig.S7f,g and R1:3/4.

We made these double KO mice for an inducible rescue of Kindlin3 KO phenotype and optimized the timing of gene excision for this very purpose. We believe that the separate TGFβ KO analysis will require in-depth analysis of microglial functions that are beyond the scope of the phenotype we have described in this manuscript.

5. As reported earlier (Xu et al. ATVB, 2014), Kindlin3 protein levels in K3KI mutant mice were equal to those in WT mice, whereas in the current study the authors report decreased mRNA levels in microglia of K3KI mice compared to WT or K3KI-Flp mice (Fig. 2h). It should be clarified if the nomenclature is the same between the two studies. In addition, Kindlin3 protein levels should be analyzed in the mutant mice.

The K3KI mutant mice reported in Xu et al. (ATVB, 2014) are referred to as K3KI-Flp mice in our study. The Kindlin3 mutant knock-in mice (K3KI) were generated by site-directed mutagenesis in exon 14 of the Kindlin3 gene, giving rise to the mutant Kindlin3 protein with Q597W598 to AA substitution. The Neo cassette present in intron 13 of the Kindlin3 gene resulted in very low levels of mutant Kindlin3 expression (referred to as K3KI in our manuscript). These mice in comparison to K3KI-flp were instrumental to show that it is Kindlin3 expression, but not mutation, that is responsible for vascular phenotype. To remove FRT-flanked neomycin

resistance gene, the K3KI mice were crossed with FLP1-expressing mice (from Jackson laboratory) for 4 generations to obtain the K3KI-flp mice that are homozygous mice for FLP1 genes and K3 mutant genes. Removal of the Neo cassette increased the expression levels of mutant Kindlin3 to levels similar to endogenous Kindlin3 of WT controls. Western blot analysis showing protein levels in primary microglia (R1:5a) and confirmation with BMDM (R1:5b) cells have now been included into Fig.S1c. We have modified methods as well to include these clarifications.

6. It has been previously reported that the K3KI mice have significantly increased numbers of T cells and slightly increased numbers of red blood cells (Xu et al., *ATVB*, 2014). Considering that in the current study the authors used K3KI mice with significantly reduced levels of Kindlin3 mRNA, the authors should discuss the potential involvement of other cells types.

As noted by the reviewer, the K3KI mice used in our study showed increased levels of WBC and a decrease in RBC as shown in Fig. R1:6, and this could potentially affect the vasculature. Hence, to confirm that blood cell levels are not the reason for abnormal vasculature in K3KI, in parallel we had generated the tamoxifen-inducible CX3CR1-driven knockout of K3, which is widely used for microglia studies. The knockout of K3 in CX3CR1-cre;K3f/f postnatal pups by tamoxifen recapitulated the phenotype of K3KI, confirming a role for Kindlin3 in microglia.

The K3KI-flp mice in our study (i.e. K3KI mice in Xu et al., *ATVB*, 2014), which express normal levels of K3 but are defective in integrin binding, did not show the abnormal vascular phenotype despite having high levels of WBC and low RBC counts. This further confirms our conclusions. Blood counts are shown in Fig. R1:6.

7. It is not clear if the vascular phenotype is transient or persistent beyond P16 in CX3CR1-cre;K3f/f and K3KI mice.

We analyzed the main phenotype in depth using additional time points as suggested by the reviewer. At the age of P21, while the number of neovascular sprouts growing from the deep vascular plexus into the ONL was decreased, all of the remaining vascular sprouts were transformed into large bulbous blood vessels resembling vascular malformations, which penetrated beyond photoreceptors. Some of these malformations merged together to form lesions under photoreceptors, resembling a fourth vascular layer that is never seen in normal mice (Fig.4g and R1:7a). These blood vessels typically replace photoreceptors and the ONL is raised as shown in Fig.R1:7a, creating a risk of its detachment from RPE. This is a very significant defect, which might lead to vascular rupture. Importantly, in Kindlin3-deficient humans with severe bleeding, this pathology might be further exacerbated.

By P60, only a certain percentage of the K3-deficient mice retained these neovascular lesions (one out of six mice; Fig.S6 e and R1:7b). The higher vascular density, however,

remained. This is a rather common phenomenon in mouse retinal models, where abnormal vasculature partially corrects itself by adulthood.

8. The difference in tissue stiffness between INL and ONL was 3-fold, whereas a 10-fold difference in substrate stiffness was used to polarize microglia in vitro (Fig. 2a,b). In addition, the stiffness experienced by microglia was much higher in vivo than in vitro. The authors should discuss these differences in the experimental conditions between in vivo and in vitro.

To isolate the effect of cell polarization alone on TGF β 1 expression while also replicating the microenvironment of microglia, we used well-characterized hyaluronic acid gels with fibronectin. Hyaluronic acid (HA), the simplest glycosaminoglycan and a major constituent of the extracellular matrix (ECM), closely resembles the mouse retinal microglial microenvironment. HA regulates cell adhesion and motility and mediates cell proliferation and differentiation, making it not only a structural component of tissues, but also an active signaling molecule. Most importantly,

microglia, which are rather sensitive to cell culture conditions, tolerates these gels very well as judged by cell survival.

Due to the limitations of commercially-available HA gel kits, we could only achieve a maximum stiffness of ~600pa, which is similar to the stiffness of OPL in retina or that of brain. The reviewer is correct that an HA gel of 600pa is relatively soft and falls at the lower end of the physiological tissue stiffness spectrum (0.1kPa- 64kPa). Hence, to get a significant and consistent difference in stiffness between our (soft and stiff) gels, we used a significant, i.e. 6-10 fold (~60pa-600pa), difference in stiffness between our gels.

To address the reviewer's concern, we optimized experimental conditions for another set of silicone gels of 0.2, 0.5, and 2kPa, replicating the changes in stiffness observed in retina. Our new Fig.S2a,b and R1:8 show that TGFβ1 expression by microglia is dependent upon substrate stiffness, as previously shown with HA gels in the manuscript.

R1:8

9. Fig. 3h. The secreted TGFβ1 levels are measured by ELISA after 24h and 48h of culture. Why the TGFβ1 levels are not increasing over time in WT cultures?

Based on our observation, WT microglia express and secrete TGFβ1, as they gradually change their morphology from round to polarized cells over a period of 24hrs. Once microglia are completely adhered, spread, and polarized, TGFβ1 expression is shut down nearly completely. Thus, from 24 to 48 hours there is no substantial increase in TGFβ1 present in media. In contrast, as we show in Fig.S5 g, K3-deficient microglia were unable to spread/polarize over a period of 48hrs, thereby constitutively producing high levels of TGFβ1, which accumulated in the media.

10. Fig. 4c. Quantification of the pSMAD3+ lectin+ cells in K3KI and WT mice would strengthen the conclusions.

As suggested, quantifications of the pSMAD3+ lectin+ cells in K3KI and WT mice have been included in Fig.4b and R1:10. We observe a 4-fold increase in the numbers of pSMAD3+ endothelial cells in K3-deficient mice as compared to WT.

R1:10

11. Fig. 7a, d. How many cells per experiment were quantified, when the localization of pMLC and pERK were analyzed? Are the differences statistically significant?

The experiments were repeated five times for pMLC (n=35 cells per group) and three times for pERK (n=40-50 cells per group) and cells were quantified. The differences are statistically significant and these details have been included in the figures.

Minor comments:

1. Fig. 3e-g. The labels should be standardized. In Fig. 3e the gene of interest is on a y-axis, in Fig. 3f, g the gene of interest is above the diagram. Corrected
2. Fig. 6b-c. The color coding of the bars should be included in the figure. Corrected
3. Fig. 6f. The loops should be quantified per 500 μm^2 as in previous figures for easier comparison. Corrected
4. A typo in row 519 should be corrected, polaization to polarization. Corrected
5. A typo in row 808 should be corrected, VEGEFA to VEGFA. Corrected
6. The authors might consider some of the earlier studies where the effects of TGF β 1 on microglia quiescence and ramification were investigated (Abutbul, Glia, 2012, Schilling, Eur J Neuroscience, 2001). Included in the discussion.

Reviewer #2:

The manuscript by Dudiki and colleagues investigate the effect of tissue stiffness in the cellular and molecular responses of microglia and their consequences to the process of neovascularization. The authors found that microglia respond to tissue stiffness in the retina and that this requires kindlin3. Furthermore, they demonstrate that deletion in K3 or inactivation of the K3KI domain results in increased TGF- β secretion and pSMAD in the endothelium. At the tissue level, the retinal vascular responds by increasing sprouting and promoting hypergrowth. Overall the study brings to light a previously unexpected role of kindlins, namely in their ability to regulate TGF- β with extremely interesting consequences to the cellular behavior of microglia and endothelial cells. The authors provide clean and stunning images, furthermore the rescue experiments offer further support to the main conclusions and bring the story to a very nice closure. However, there are several aspects that deserve further evaluation, in particular, some controls are missing, editorial (or experimental) changes should be made to ensure that the stated conclusions remain true to the findings.

We thank the reviewer for the constructive comments and questions, which are addressed in detail below.

1. The stated conclusion that: “the microglia mechanosensing is independent of integrins” has not been fully supported by the data. What the authors show is that: (a) Kindlin3’s effect on integrins is not required for mechanosensing, instead the needed region is the KI domain of this intracellular adaptor protein and that (b) beta 1 and 2 integrins are not required (although controls for these KOs are needed – see below) but the authors did not evaluate all integrins. In order to state that integrins are not part of this mechanism, additional experiments are needed, as well as a comprehensive evaluation of each integrin expressed by microglia, which expand beyond the beta 1 and beta 2 families. Alternatively, this reviewer would suggest to revise the statement to bring more specificity and alignment with what has been shown, instead of making a broad generalization that does not reflect the experiments included in the study.

In addition to Kindlin3 KO in microglia, we analyzed Kindlin3 Knock-in mice bearing a mutation disrupting Kindlin3-integrin (including all integrins, not limited to β 2 and β 1 integrins) interactions (K3KI-flip with normal Kindlin3 expression). Also, we analyzed these same Kindlin3 knock-in mice with low Kindlin3 expression (K3KI). The vascular misdirection was only present in mice where Kindlin3 expression was altered, whereas in mutants where Kindlin was present but lacked the integrin binding site (K3KI-flp), vasculature was normal. Thus, based upon these three Kindlin mouse lines, we concluded that microglial Kindlin3-integrin interactions are not required for neovascular growths into ONL. The wording was changed to avoid any confusion, since integrins are crucial for several aspects of TGF β signaling, including its activation within the extracellular matrix.

The statement was revised to emphasize this point. The results are shown in Fig.5 and are discussed in lines 312-313. Likewise, individual integrin β 1 and β 2 knockout mice did not exhibit vascular overgrowth into ONL.

We have also included in-depth characterization of β 1 integrin KO mice. While there was a significant increase in the number of vascular loops, vascular growths into the ONL that are

characteristic of TGF β overexpression were not present. Also, while there were morphological changes in microglia, their polarization was not as drastically affected as in K3KI or K3KO (Fig.2g and R1:2d,e,f). *In vitro*, β 1 integrin KO microglia showed a small decrease in polarization (Fig.S1g). However, in contrast to K3KO, TGF β 1 expression in β 1 KO microglia was not affected (Fig. R1:2g). Microglia from β 2 KO mice did not exhibit any changes in morphology either *in vivo* or *in vitro*. Moreover, we performed western blotting of retinal lysates to show that pSMAD3 levels in β 1 integrin knockout mice were comparable to WT, whereas there was a dramatic increase in both K3KO and K3KI (but not in K3KI-flp with normal Kindlin3 expression of integrin-deficient mutant). This is shown in new fig.S6a.

2. Along the lines with #1, the statement “microglia respond to tissue stiffness in integrin-independent fashion” needs editorial clarity and needs to be better controlled. The authors must provide evidence that in fact all the KOs shown in panel “f” had absence or significant reduction of the protein being targeted by Cre at the genomic level.

As requested, western blot analysis of primary microglia or bone marrow-derived macrophage (BMDM) lysates for the levels/absence of K3, as well as β 1 and β 2 integrins, in their respective mouse models was performed. K3 levels in microglia were undetectable in the K3-deficient models: K3KI and tamoxifen-treated K3KO mice. K3KI-flp knock-in mice as discussed above express Kindlin levels comparable to WT. GAPDH was used as loading control (Fig.S1c and R2:2).

β 1 integrin was also efficiently knocked out in B1KO mice upon tamoxifen treatment. β 2 global knockout mice were purchased from Jax labs and are well characterized. These mice show very low levels of β 2 integrin in retina. GAPDH was used as loading control (Fig.S1d,e and R2:2).

3. The authors have convincingly shown that TGF-beta levels change by two to three-fold in the K3KI microglia. While this is impressive, it is unclear to what extent this impacts signaling. For example, *in vitro*, addition of 1 ng/ml or 10 ng/ml of TGF-beta have a similar outcome in activation of Smad3 and 4. This is because the rate-limiting constituents are the receptors, rather than the ligand in this signaling cascade *in vitro* (this is not the case for VEGF, however). Can the authors assess whether this two-three fold change in TGF-beta is sufficient to translate into a Smad3/4 signaling? This reviewer appreciates the results shown in Figure 4a, however, this does not directly prove that the differences in pSMAD3 relate to TGF-beta, instead, it is possible that levels of SMAD3 or TGF-beta receptors might be distinct in K3KI microglia (a comprehensive transcriptional profile would be important – as also stated below).

We completely agree with the presented logic. As stated by the reviewer, K3KI microglia express two- to three-fold higher levels of TGFβ1 and this increase clearly translates to increased phospho-SMAD3 levels. To clarify this point, we added a separate quantification of SMAD3/4 phosphorylation in endothelial cells, which is a main target for microglial TGFβ. As shown in Fig. 4b, an increase in TGFβ translated into a ~5 fold increase in pSMAD-positive endothelial cells *in vivo*. Neither TGFβ receptors nor SMAD expression in endothelium were affected by the lack of Kindlin3, which is microglia-specific as documented in Fig. R2:4.

The role of microglial TGFβ in the observed phenotype was further shown using a microglia-specific Kindlin KO, where upon tamoxifen treatment K3 is knocked out only in the microglia, but no other cell types in the retina. This microglia-specific K3 knockout model replicated the phenotype of K3KI, and together with microglia depletion experiments showed that vascular overgrowth was affected by microglial TGFβ1. In line with these models, we show rescue by inducible microglia-specific double knockouts of Kindlin3 and TGFβ together. SMAD signaling activation in K3KO and K3KI, but not in WT and K3KI-flp, was further verified by western blotting of retinas (see new Fig. S6a, R2:5, and discussion below).

To further address the reviewer's concern, we performed microglia-endothelial cell co-culture studies. As shown in Fig. S5g, K3KI microglia *in vitro* secrete two- to three-fold higher levels of TGFβ1 into the media. This increase in TGFβ1 was sufficient to increase pSMAD3 levels in RF6A endothelial cells (monkey choroid-retina endothelial cell line) as shown in Fig. S5h,i and R2:3. The percentage of nucleus-specific pSMAD3 staining in RF6A line increased by more than 2 folds in the presence of K3KI microglia as compared to WT. We believe that the combination of multifaceted *in vivo*, *ex vivo*, and *in vitro* approaches conclusively supports our conclusions on the role of microglia-derived TGFβ1.

R2:3

4. While the selection of TGF-beta was logical and the authors have also shown TNFalpha, VEGF, sflt 1 and IL1beta, all of these have been hand-picked. Considering the availability of technology today, it will be important to comprehensively profile WT and Kindlin3-KO microglia and determine if indeed these are the key factors that affect EC biology. Such unbiased approach is important to offer support to the conclusions put forward.

None of these targets were in fact “hand-picked”. First, we had in hand results of RNAseq analysis of entire retinas from K3KI and WT mice, which showed changes in 140 genes from sensory perception to eye development (see Fig.S4a and R2:4), possibly reflecting the consequences of this pathology on photoreceptors. Unfortunately, there were no consistent changes in microglia-specific growth factors or cytokines, possibly due to the low proportion of these cells as compared to photoreceptors. Next, with microglia in mind, we analyzed our old gene arrays of myeloid cells from Kindlin3-deficient humans. These arrays showed altered VEGF, IL-10, and TNF expression (Supplementary Tables 1,2 and Fig. R2:4). All of these targets were verified; however, no differences between WT and K3KO microglia were found at the protein level.

Together, as can be seen in Fig.S4 and Fig.S5, we analyzed a number of secreted factors by ELISA-based assays and no differences were found between WT, K3KO, or K3KI microglia except TGFβ1.

As an additional guide, we used the published results of large-scale proteomics analysis of meta-adhesome, i.e. protein complexes assembled as a result of adhesion to fibronectin (~2,000 proteins) (See “*Definition of a consensus integrin adhesome and its dynamics during adhesion complex assembly and disassembly*” by Horton et al., *Nat Cell Biol*, 2015). Importantly, in these sets, our target Kindlin3 exhibits a first degree of interaction. We tested every interleukin and growth factor enriched within these complexes, with an emphasis on proteins showing changes during adhesome assembly and disassembly (Supplementary Tables 11 and 12), including TNFα and TGFβ. Remarkably, TGFβ1 was the only secreted factor showing a trend opposite to Kindlin3 (FERMT3), which is in agreement with our results.

However, we believe that it is daunting to explain all of the details of our search process. Thus, we show the tables of altered pathways in entire retinas, ELISA-based results of microglia-secreted factors, and include citations of appropriate manuscripts.

R2:4

a Leukocytes from Kindlin3 deficient patients vs control:

Table 1: List of vascular genes, changed genes are in red

Gene name			Species	Change
ATPase copper transporting alpha(ATP7A)	ATPase copper transporting alpha(ATP7A)	RG	Homo sapiens	
ELK3, ETS transcription factor(ELK3)	ELK3, ETS transcription factor(ELK3)	RG	Homo sapiens	
G protein-coupled estrogen receptor 1(GPER1)	G protein-coupled estrogen receptor 1(GPER1)	RG	Homo sapiens	
Jun proto-oncogene, AP-1 transcription factor subunit(JUN)	Jun proto-oncogene, AP-1 transcription factor subunit(JUN)	RG	Homo sapiens	
RAR related orphan receptor A(RORA)	RAR related orphan receptor A(RORA)	RG	Homo sapiens	
Rap guanine nucleotide exchange factor 2(RAPGEF2)	Rap guanine nucleotide exchange factor 2(RAPGEF2)	RG	Homo sapiens	
SRY-box 4(SOX4)	SRY-box 4(SOX4)	RG	Homo sapiens	
StAR related lipid transfer domain containing 13(STARD13)	StAR related lipid transfer domain containing 13(STARD13)	RG	Homo sapiens	
T brachyury transcription factor(T)	T brachyury transcription factor(T)	RG	Homo sapiens	
TCDD inducible poly(ADP-ribose) polymerase(TIPARP)	TCDD inducible poly(ADP-ribose) polymerase(TIPARP)	RG	Homo sapiens	
TNFSF12-TNFSF13 readthrough(TNFSF12-TNFSF13)	TNFSF12-TNFSF13 readthrough(TNFSF12-TNFSF13)	RG	Homo sapiens	
Wnt family member 5A(WNT5A)	Wnt family member 5A(WNT5A)	RG	Homo sapiens	
adrenomedullin(ADM)	adrenomedullin(ADM)	RG	Homo sapiens	
annexin A3(ANXA3)	annexin A3(ANXA3)	RG	Homo sapiens	
aryl hydrocarbon receptor(AHR)	aryl hydrocarbon receptor(AHR)	RG	Homo sapiens	
atypical chemokine receptor 3(ACKR3)	atypical chemokine receptor 3(ACKR3)	RG	Homo sapiens	
beta-1,4-galactosyltransferase 1(B4GALT1)	beta-1,4-galactosyltransferase 1(B4GALT1)	RG	Homo sapiens	
cadherin 2(CDH2)	cadherin 2(CDH2)	RG	Homo sapiens	
calcitonin receptor like receptor(CALCRL)	calcitonin receptor like receptor(CALCRL)	RG	Homo sapiens	
caveolin 1(CAV1)	caveolin 1(CAV1)	RG	Homo sapiens	
collagen type I alpha 1 chain(COL1A1)	collagen type I alpha 1 chain(COL1A1)	RG	Homo sapiens	
cystathionine gamma-lyase(CTH)	cystathionine gamma-lyase(CTH)	RG	Homo sapiens	
cytochrome P450 family 1 subfamily B member 1(CYP1B1)	cytochrome P450 family 1 subfamily B member 1(CYP1B1)	RG	Homo sapiens	
dimethylarginine dimethylaminohydrolase 1(DDAH1)	dimethylarginine dimethylaminohydrolase 1(DDAH1)	RG	Homo sapiens	
eukaryotic translation initiation factor 2 alpha kinase 3(EIF2AK3)	eukaryotic translation initiation factor 2 alpha kinase 3(EIF2AK3)	RG	Homo sapiens	
fibroblast growth factor 9(FGF9)	fibroblast growth factor 9(FGF9)	RG	Homo sapiens	Decrease
fibroblast growth factor receptor 1(FGFR1)	fibroblast growth factor receptor 1(FGFR1)	RG	Homo sapiens	
fibronectin 1(FN1)	fibronectin 1(FN1)	RG	Homo sapiens	
forkhead box O1(FOXO1)	forkhead box O1(FOXO1)	RG	Homo sapiens	
frizzled class receptor 5(FZD5)	frizzled class receptor 5(FZD5)	RG	Homo sapiens	
heart development protein with EGF like domains 1(HEG1)	heart development protein with EGF like domains 1(HEG1)	RG	Homo sapiens	
integrin subunit beta 3(ITGB3)	integrin subunit beta 3(ITGB3)	RG	Homo sapiens	
interleukin 1 alpha(IL1A)	interleukin 1 alpha(IL1A)	RG	Homo sapiens	
lymphoid enhancer binding factor 1(LEF1)	lymphoid enhancer binding factor 1(LEF1)	RG	Homo sapiens	
myosin heavy chain 10(MYH10)	myosin heavy chain 10(MYH10)	RG	Homo sapiens	

Gene name			Species	Change
neurotrophic receptor tyrosine kinase 2(NTRK2)	neurotrophic receptor tyrosine kinase 2(NTRK2)	RG	Homo sapiens	
notch 1(NOTCH1)	notch 1(NOTCH1)	RG	Homo sapiens	
nuclear receptor subfamily 2 group F member 2(NR2F2)	nuclear receptor subfamily 2 group F member 2(NR2F2)	RG	Homo sapiens	
phospholipid phosphatase 3(PLPP3)	phospholipid phosphatase 3(PLPP3)	RG	Homo sapiens	
platelet derived growth factor subunit A(PDGFA)	platelet derived growth factor subunit A(PDGFA)	RG	Homo sapiens	
protein kinase C beta(PRKCB)	protein kinase C beta(PRKCB)	RG	Homo sapiens	
protein kinase, X-linked(PRKX)	protein kinase, X-linked(PRKX)	RG	Homo sapiens	
protein tyrosine phosphatase, non-receptor type 14(PTPN14)	protein tyrosine phosphatase, non-receptor type 14(PTPN14)	RG	Homo sapiens	
regulator of cell cycle(RGCC)	regulator of cell cycle(RGCC)	RG	Homo sapiens	
relaxin 2(RLN2)	relaxin 2(RLN2)	RG	Homo sapiens	
reversion inducing cysteine rich protein with kazal motifs(RECK)	reversion inducing cysteine rich protein with kazal motifs(RECK)	RG	Homo sapiens	
roundabout guidance receptor 1(ROBO1)	roundabout guidance receptor 1(ROBO1)	RG	Homo sapiens	
semaphorin 4A(SEMA4A)	semaphorin 4A(SEMA4A)	RG	Homo sapiens	
signal transducer and activator of transcription 1(STAT1)	signal transducer and activator of transcription 1(STAT1)	RG	Homo sapiens	
spermidine/spermine N1-acetyltransferase 1(SAT1)	spermidine/spermine N1-acetyltransferase 1(SAT1)	RG	Homo sapiens	
tetraspanin 12(TSPAN12)	tetraspanin 12(TSPAN12)	RG	Homo sapiens	
transforming growth factor beta receptor 2(TGFB2)	transforming growth factor beta receptor 2(TGFB2)	RG	Homo sapiens	
vascular endothelial growth factor A(VEGFA)	vascular endothelial growth factor A(VEGFA)	RG	Homo sapiens	Decrease

Table 2: List of genes in cytokine response category, changed genes are in red

Gene Name	Gene Name	Gene Name
Jun proto-oncogene, AP-1 transcription factor subunit(JUN)	interferon gamma(IFNG)	interleukin 18 receptor accessory protein(IL18RAP)
MAF bZIP transcription factor(MAF)	interleukin 1 alpha(IL1A)	interleukin 18(IL18)
RAR related orphan receptor A(RORA)	interleukin 1 beta(IL1B)	interleukin 2 receptor subunit gamma(IL2RG)
RAR related orphan receptor C(RORC)	interleukin 10(IL10)	interleukin 2(IL2)
RELA proto-oncogene, NF-kB subunit(RELA)	interleukin 12 receptor subunit beta 1(IL12RB1)	interleukin 21 receptor(IL21R)
SMAD family member 2(SMAD2)	interleukin 12 receptor subunit beta 2(IL12RB2)	interleukin 21(IL21)
SMAD family member 3(SMAD3)	interleukin 12A(IL12A)	interleukin 22(IL22)
T-box 21(TBX21)	interleukin 12B(IL12B)	interleukin 23 receptor(IL23R)
forkhead box P3(FOXP3)	interleukin 13(IL13)	interleukin 23 subunit alpha(IL23A)
interferon gamma receptor 1(IFNGR1)	interleukin 17A(IL17A)	interleukin 4 receptor(IL4R)
interferon gamma receptor 2 (interferon gamma transducer 1)(IFNGR2)	interleukin 17F(IL17F)	interleukin 4(IL4)
	interleukin 18 receptor 1(IL18R1)	interleukin 5(IL5)

b Retina gene array pathway analysis:

c UPLEX, ELISA based Assay of microglia lysates:

5. Controls for Figure 6 – while the levels of pSMAD3 are impressive in the image provided, imaging for assessment of levels is not the most reliable approach. Instead, Western blots showing total SMAD3/ pSMAD3, will unequivocally support the statements put forward and contribute to address some of the previously raised concerns. As discussed previously in relation to integrins, the authors must provide assessment of the level of the Cre deletion on protein (for TGF-b in the case of figure 6).

As suggested by the reviewer, we have performed western blot analysis of retinas for pSMAD3/SMAD3 levels (Fig.S6a and R2:5). Shown are two different western blots probed with antibodies for pSMAD3 and SMAD3, followed by β -actin for a loading control. The pSMAD3/SMAD3 levels were determined after adjusting SMAD3 levels to the actin levels. The K3KI and K3KO retinas show increased pSMAD3 levels compared to WT. At the same time, K3KI-flp knock-in of Kindlin3 lacking integrin interaction was similar to WT. This further supported our imaging quantification in Fig.6. A significant reduction in pSMAD3/SMAD3 levels was observed for K3KO/TGFB1 KO retinas, indicating efficient deletion of microglial TGFB1. Importantly, pSMAD3 levels in β 1 microglia-specific KO were similar to those in WT.

R2:5

6. The experiments presented in Figure 7 contribute to address a key question in the manuscript, namely: how could kindlin3 regulate TGF-beta1's secretion in microglia. Using blebbistatin, the authors normalized the hypercontraction of microglia and the levels of TGF-beta. Were the transcriptional levels of TGF-beta also changed/normalized? (Figure 3e shows a 3.5 fold increase in K3KI).

Also in this figure, it is puzzling that the levels of secreted TGF-beta in Figure 3h are 2 to 3 fold increased in K3KI, while in Figure 7 this does not reach 1.5 fold (average). Can the author comment on this discrepancy?

We used blebbistatin to normalize the hypercontraction of K3KI microglia, which resulted in downregulation/normalization of TGFβ1 transcription as assessed by qPCR (Fig.7). Fig.3h (now moved to Fig.S5g) is a representative ELISA for secreted TGFβ1 levels, while Fig.7 shows mRNA levels of TGFβ1 upon blebbistatin treatment. The possible explanations for the differences in fold changes between the two experiments include:

Based on our experience, microglia are very sensitive to *in vitro* conditions but are most stable when kept as a mixed glial cell culture. Hence, the freshly-harvested microglia from this mixed culture consistently showed the most dramatic differences in TGFβ1 mRNA levels between WT and K3-deficient microglia (2-4 fold, Fig.3e of manuscript).

For secreted TGFβ1 at the protein level, microglia were cultured in serum-free media (there is a substantial amount of TGFβ coming from aggregated platelets, which are the main source of TGFβ in blood). The change of media alone might account for the differences between RNA and protein levels in those experiments. After 24 hrs, we observed a 2.5-fold increase in TGFβ1 protein, which accumulated over this time.

For blebbistatin treatment, control cells were treated with levels of DMSO in media (a vehicle for blebbistatin) for 12 hrs. This could have had some effect on the microglial cell conditions and gene expression, including TGFβ1.

To address the reviewer's concern of heterogeneity in TGFβ1 expression, we used two independent CRISPR-K3KO clones of myeloid RAW cell lines (K3KO1 and 2), which exhibited a lack of spreading (Fig.7g in manuscript). Similar to K3KO microglia, K3KO RAW cells express higher levels of TGFβ1. Blebbistatin treatment of K3KO RAW cells normalized their hypercontractility, resulting in downregulation/normalization of TGFβ1 transcript levels as analyzed by qPCR (Fig.S8d-f and R2:6).

7. In several occasions, the authors mention retinal angiomatous proliferation (RAP), the impact of the present manuscript would significantly benefit from comparing the vasculature pattern of K3KI mice to RAP specimens. Also, Are the levels of TGF-beta comparable? or What is the degree of pSMAD3 in the endothelium?

There are only three other known models for RAP: *Vldlr* knockout, VEGF-overexpressing transgenic, and JR5558 (NRV2) mice. In all of these models, the vascular abnormality is dependent upon overexpression of VEGF by photoreceptors, independent of microglia. We show the very first study where microglial deficiency causes a dramatic vascular phenotype. Analysis of vasculature in K3KO mice at later stages revealed substantial vascular malformations under the layer of photoreceptors, forming a fourth vascular layer and dramatically thickening the retina (new Fig.4g and R1:7). Considering that human Kindlin3-deficient patients are prone to bleeding, these malformations represent a substantial risk of retinal damage and detachment.

Minor Concerns:

In the summary, the opening statement “Tissue mechanics regulate development and virtually all pathologies of the Central Nervous System (CNS) by largely unknown mechanisms” is a bit too bold. While it is clear physical forces are extremely important in biological tissues, the role of tissue mechanics in development and pathology of the CNS

has not been sufficiently investigated to make such statement. Some might even say that Alzheimer's Disease for example, a rather prevalent pathology of the CNS does not require any contribution by tissue mechanics for the onset of the disease – despite the fact that at later stages the presence of plaque, as stated by authors, thickness of retina might later (as a consequence of the disease) alter physical stiffness of the neural tissue. In any case, there is no need to start with a statement that might turn investigators away from reading the rest of the paper. I strongly suggest re-writing.

As suggested, we have rewritten the statement as follows: Tissue mechanics play an important role in development and pathologies of the Central Nervous System (CNS) by largely unknown mechanisms.

- In Figure 4c, the use of colors (red and magenta) are unfortunate. To better see colocalization, pls. do change red to green or blue. This should be trivial with the original images.

As suggested, the colors in Fig.4c (now 4e) and 2d have been changed for better visualization.

Reviewer #3:

The manuscript by Dudiki et al. aimed to investigate microglial mechanosensing and its role in regulating vascular patterning in the retina. The authors conclude that microglia migrate through the tissue stiffness gradient, change their morphology, and subsequently regulate vascularization through a Kindlin3 and subsequent TGF β 1 signaling mechanism. The idea that microglial morphology and function is regulated by a mechanosensing mechanism is interesting and novel and elucidating microglial functions in vascular development is also important. However, there are major concerns with this paper regarding whether the data support the major conclusions and the connection between microglial mechanosensing and vascular regulation is weak. These and other concerns are outlined below:

We thank the reviewer for the constructive comments and questions which are addressed in detail below.

1.) The conclusion that microglia change morphology based on tissue stiffness in vivo is not convincing. The authors show correlative changes in microglial morphology within tissue with different stiffnesses. They also showed in vitro that modulating the substrate stiffness can affect microglial morphology. However, all these data are correlative and do not provide sufficient evidence in vivo that tissue stiffness directly regulates microglial morphology. In vitro systems are quite different than in vivo settings and are not sufficient to support the author's claim in vivo. The authors would need to modify tissue stiffness in vivo and show that it directly regulates microglial morphology through microglia-specific receptors.

Technical approaches allowing changing tissue stiffness *in vivo* are limited to nonexistent and have not been established in retinal tissues. We decided to closely follow the recent pioneering studies by the Franze group, which take an advantage of a *Xenopus* model where larvae's exposed brains are treated with chondroitin sulfate solution to make the tissue softer. We adapted this technology for live retinas. Several parameters were controlled in this model: first, we demonstrated that chondroitin sulfate at a concentration of 20mg/ml actually altered the stiffness of retinas using atomic force microscopy. Indeed, we were able to detect a nearly 2-fold decrease in Young's modulus of treated retinas, which corresponded to the change from ~900pa in control to 400pa (as shown in new Fig.1g and in R3:1a). We focused on the area surrounding the optic nerve (~700-1200 μm) since most of the vascular malformations are observed in that particular region. We also show the heterogeneity of retinal stiffness in control and chondroitin sulfate-treated retinas. The distribution is shown in Fig R3:1b,c. The shift towards lower stiffness upon chondroitin sulfate treatment is obvious. The second important parameter is microglial status, which was monitored by live imaging. We were able to observe microglial responses up to 6 hours after initiation of the treatment without a substantial effect on microglial viability as judged by process movements. Upon chondroitin sulfate treatment, a substantial proportion of WT microglia lost their alignment and polarization (Fig.1h,l and R3;1f). The percentage of polarized microglia in WT retinas decreased from ~85% to 40%, closely resembling the polarization status of control K3KI microglia (Fig.3f,g and R3:1h,i). The microglial polarity index also decreased from 4 to 2.5 upon "softening" of WT retinas (Fig.1f and 1i). Importantly, chondroitin sulfate treatment, and changes in stiffness associated with it, did not affect polarization of K3KI microglia as shown in new Fig.3f,g, consistent with our *in vivo* and *in vitro* studies and further supporting our conclusion that mechanosensory function is impaired by the lack of Kindlin3.

The overall conclusion that microglia respond to retinal tissue stiffness is made not only based upon WT retina analyses and *in vitro* experiments, but also importantly on the analysis of Kindlin3-deficient retinas, where microglia lack mechanosensing function as shown in detail both *in vivo* and *in vitro*. We completely agree that *in vitro* systems only partially recapitulate *in vivo* situations. At the same time, precise manipulations with *in vivo* parameters such as stiffness and matrix composition in deep tissues might not be specific or even technically possible. In most cases, the use of a mutant that impairs the response to stiffness without affecting other microglial processes represents the best available tool to establish a strong causative connection. We would like to note that the phenotypes and microglial responses *in vivo* closely resembled our results obtained in HA and silicone gels, also supporting the appropriateness of those models.

Biomechanical Characterization of Live Retinal Tissue Control vs Chondroitin sulfate (treated)

2.) The authors propose that microglial mechanosensing occurs through Kindlin 3, but they provide no evidence that this is a mechanosensing receptor. More work is required to show Kindlin 3 is a mechanosensing cell adhesion molecule and it would be important to determine the extracellular ligand. It is unclear what is binding Kindlin3.

According to the reviewer's suggestion, we have further strengthened this conclusion that the lack of Kindlin impairs mechanosensing of microglia. In our new experiments described above and shown in new Fig.1g,h,i, and R3:1, retinal stiffness was altered by chondroitin sulfate

treatment. This resulted in a loss of alignment and polarization of WT microglia, but had no effect on polarization index in K3KI microglia as shown in new Fig.3f,g. This establishes that tissue stiffness changes are not sensed by Kindlin3-deficient microglia. The same is shown in numerous models using substrates of various stiffness (both HA- and silicone gel-based) as shown in Fig.2j,k and I and Fig.S2 a,b. Of note, silicon gel stiffness closely resembles retinal stiffness.

To even further strengthen support for the role of Kindlin3, we knocked it out by CRISPR in an alternative myeloid cell model and show their lack of polarization, as well as the lack of TGF β adjustment.

Kindlin 3 is an intracellular integrin binding adaptor molecule containing 3 FERM (Fermitin-Ezrin-Radixin-Moesin) domains and a PH domain (shown in the diagram in Fig.8 and R3:2); its alternative name is FERMT. It is structurally similar to other proteins such as ezrin that are proposed to define mechanosensory functions.

R3:2

3.) The Kindlin 3 mutant mice potentially have a very interesting microglial phenotype, but this does not seem to be specific to the cell polarity they measure or mechanosensing function. In addition to changes in cell polarity, the microglial cells in these mutant mice appear to have significant changes in their branch numbers, soma size, and cell numbers. The authors state that cell numbers were no different in the mutant mice, but this does not seem to be the case in the images they provide and they provide no quantification of these data. This requires more assessment of microglial phenotypes and the role for Kindlin 3 in modulating these phenotypes, which are less likely due to changes in mechanosensing. It would also be helpful to assess microglia morphology in these mutants more globally in the retina and in the CNS.

As per the reviewer's suggestion, we now show a detailed characterization of K3-deficient microglial morphology from retina and brain (Fig.S3 b-g and R3:3). Similar to retina, K3KI

microglia from brain show a lack of polarity, somewhat shorter filopodia, increased soma volume, and reduced branching of processes. All of these parameters, especially cell polarity, are characteristic of poorly-spread microglia in soft tissue or microglial irresponsive to tissue stiffness. As shown above in Fig.3f,g and R3:1, as a result of softening the tissue around WT microglia, cells assume a phenotype similar to K3 deficiency. At the same time, changes in stiffness had no effect on K3-deficient microglia in retinas. We also performed an opposite “rescue”, where Kindlin-3-deficient microglia were polarized by altering their cytoskeletal contractility. This led to the reversal of TGF β expression. Similar experiments were performed using Kindlin3 Crispr-based knockouts in an alternative myeloid cell line, where forced polarization of Kindlin3-deficient cells diminished and corrected TGF β levels. Together, these multiple approaches indicate that in the absence of Kindlin3, microglia fail to sense tissue stiffness.

Quantification of microglial numbers is shown in Fig.S3a and R3:3a. There was no significant change in microglial numbers in the deep layer at P12. The small increase in K3KI microglial numbers (statistically not significant) was observed only at P16 in K3KI (Fig R3:3a). However, K3KO showed a small decrease in microglial numbers at the same age (Fig R3:3a). This indicates that any variations in microglial numbers of K3-deficient mice did not contribute to the vascular phenotype observed in these mice.

4.) The authors cannot conclude that their effects are microglia specific. Perivascular macrophages, which have a similar lineage to microglia and also express CX3CR1creER, likely also express Kindlin3 and TGF β 1. Therefore, their effects may not be microglia-specific. To better support their conclusions, the authors should provide evidence (e.g. in situ and/or immunostaining) that Kindlin 3 is restricted to only microglia in the retina.

To show that Kindlin3 is restricted to microglia in retinas, we have performed staining for two additional microglial markers that are not expressed in macrophages: Iba-1 and newly-described marker TMEM-119. As shown in Fig.S2d and R3:4a,b and d, there is 100% overlap between Cx3CR1-GFP and Iba-1 and Cx3CR1-GFP and TMEM-119. These observations are consistent with the literature showing that perivascular macrophages in retina/brain are observed in inflammation-associated pathologies upon disruption of the blood-brain barrier (BBB), but are typically absent in normal-developing retinas. Numerous studies focused on the origins of microglia have demonstrated that blood-derived circulating cells are able to populate the CNS, and therefore contribute to the microglial pool, only upon the loss of BBB integrity (Ajami et al., Mildner et al.). However, a subsequent study by Rosi et al. using a combination of parabiosis and myeloablation showed that recruited monocytes/macrophages are short-lived and do not contribute to resident microglia. The presence of Kindlin3 strictly in retinal and brain microglia is shown in Fig.S2c or R:3:4c and has also been addressed in our previous studies (Meller et al. *JCI Insight*). Altogether, we did not detect any monocytes/macrophages in our retinal models at any stage of retinal development. At present, Cx3CR1-cre is considered to be the most appropriate for microglia (Qin et al., *Cell*, 2018).

R3:4

No perivascular macrophages

K3 is microglia specific in retina

5.) There is literature that demonstrates a role for TGFβ1 signaling in regulation of the vascular (Shih et al. 2003, Walshe et al. 2009, Arnold et al. 2014). Additionally, there is also published literature showing that microglial depletion causes changes to vascular complexity (Fantin et al. 2010). Therefore, without further assessment of the Kindlin3 mechanism, the study lacks novelty.

We agree that the literature of TGFβ is substantial and that manuscripts TGFβ are published in high-profile journals nearly every month due to the complexity of TGFβ effects, its multifaceted regulation, and key roles in many if not all pathologies. Many TGFβ effects are also local and difficult to pinpoint, which is the case for microglia and CNS (Springer et al., *Cell*, 2018). As the reviewer mentioned, there are pro-survival effects of pericytes on vasculature (Shin et al., 2003), TGFβ-mediated leukocyte rolling (Walshe et al. 2009), and integrin αpha8bet 6 – dependent aspects of TGFβ activation in CNS (Arnold et al. 2014). None of these address a phenomenon even indirectly similar to our study. We are not aware of any vascular biology studies connecting the regulation of microglial TGFβ by stiffness to vascular architecture in retinas. The use of inducible microglia-specific knockout of TGFβ1 to completely eliminate excessive vasculature beyond the photoreceptors while correcting vascular density is novel as well.

6.) The signaling between Kindlin3 and TGF β 1 to regulate the vasculature requires more work. All mechanistic work assessing downstream signaling from Kindlin 3 was performed *in vitro*. It is unclear if this translates *in vivo*. Further, the authors show inconsistent data related to pMLC *in vitro* (Fig 7h) where K3KO1 shows increased pMLC, but this is not replicated in K3KO2. The levels of pMLC appear to be negatively correlated with TGF β 1 levels in the K3KOs.

We aimed to perform all critical mechanistic experiments *in vivo* using multiple mouse lines (knockouts as well as knock-ins); key rescue experiments to translate our observations back to the complexity of retinal vasculature were also performed *in vivo*. New causative *in vivo* experiments were also added as described above. To add another line of *in vivo* evidence and to translate the signaling aspects of Kindlin3 to *in vivo*, we performed western blot analysis of collected retinas to demonstrate that high levels of pERK1/2 (Fig.S8b and R3:6a,b) are persistent *in vivo*, similar to our previous *in vitro* results.

We also use two independent CRISPR Kindlin3 KO lines to reconfirm our observations from primary cells isolated from Kindlin3 knockout and knock-in lines. We repeated the experiment and now show average changes upon Kindlin3 knockout as compared to WT control (Fig.S8c and R3:6c). Although the absolute values of pMLC differ between two KO lines, on average pMLC, pERK, and TGF β expression are increased upon Kindlin3 knockout.

7.) The developmental time course of microglial morphology changes in the deep versus intermediate vascular layers requires clarification. The authors suggest that microglia morphology is regulated by tissue stiffness and that the cells on the deep layers are on the stiffest ONL. However, these cells also are in contact with the less stiff IPL and they are located within the intermediate vascular layer, which has an even lower degree of stiffness than the IPL or ONL. Also, does the stiffness of the layers change across development? If not, it is unclear why microglia would have similar cell polarity in the intermediate (less stiff) and deep (more stiff) vascular layers early in development.

We added several additional time points to show when and how microglia populate the retina (Fig.S1a,b and R3:7/8). At P6, they had finished populating the superficial layer and

migration downwards was initiated. While reaching the deeper layers, microglia experienced higher stiffness, which was reflected by the changes in spreading and polarization. Until P16 (see figures in our response 3:7/8, right panel) microglia did not reach the bottom of the deep layer, and therefore were not fully polarized. The entire change within deep microglia occurred in a time-dependent manner from ramified (polarity index 1.5 ± 0.1 at P9) to bipolar rod shape (polarity index 3.6 ± 0.3 at P16). At the developmental stage of P9, microglia had just established their first contacts with the ONL, and their entire soma were still within the softer INL/OPL layer. These microglia were still ramified. At P12, microglial spreading was just initiated, but not completed. It was not until P15-16 that these cells were completely spread and polarized on the ONL. We do not see reasons (including literature analysis) to suspect that the stiffness of the layers per se differed dramatically between P12 and P15-16.

R3:7/8

c

8.) The authors conclude that microglia are directionally migrating, but they show no time-lapse imaging to show migration. This terminology should be modified.

Since it is technically impossible to show time-lapse videos of the back of the eye for 16 days, we present nearly day-by-day snap shots of microglial migration into the retina (Fig.S1a,b and R3:7/8). Based upon this chronological retinal imaging, at P6 microglia were observed only within the superficial layer. By P9-12, the microglial numbers in the superficial layer decreased with a simultaneous increase in middle/deep layer, indicative of microglial migration into the deep layers. In addition, individual microglial morphology (cell soma extending between layers) was indicative of microglial migration. By P16, the migrating microglial morphology was not observed, but three clearly-defined layers of microglia were seen. This is consistent with published literature.

9.) The methods described for microscopy require more detail. For example, how were images acquired for a given experiment, what magnification was used, were z-stacks collected, how large were the z-steps etc.

All of the details have now been included.

10.) While n's are given in the figure legend, it is frequently difficult to determine if the authors are referring to fields of view, animals, cells, etc. There is also large spreads in the n's. For example, in some experiments the n=5-7(Fig 1d) where as in other experiments the n=36 (Fig 1f), 60 (Fig 2b) etc. As it is written, the authors say this is per group. Does this mean 36, 60, etc. mice were quantified? If this reflects cell number, this is not appropriate and statistics should be re-run with n's defined as animals. If this is animal number, it is unclear why so many mice were used for one experiment and not others.

All of the details have now been included.

11.) The statistics run in this manuscript appear to be incorrect. The authors state that used an unpaired t-test as their posthoc analyses following ANOVAs. The statistics should be re-run and a posthoc analysis such as Tukey's or Bonferroni's should be used.

Statistics were re-run according to the reviewer's suggestion.

12.) There are many grammatical errors throughout, which makes the manuscript a difficult read. This requires significant editing.

Corrected.

References:

- 1 Shih, S. C. *et al.* Transforming growth factor beta1 induction of vascular endothelial growth factor receptor 1: mechanism of pericyte-induced vascular survival in vivo. *Proc Natl Acad Sci U S A* **100**, 15859-15864, doi:10.1073/pnas.2136855100 (2003).
- 2 Allinson, K. R., Lee, H. S., Fruttiger, M., McCarty, J. H. & Arthur, H. M. Endothelial expression of TGFbeta type II receptor is required to maintain vascular integrity during postnatal development of the central nervous system. *PLoS One* **7**, e39336, doi:10.1371/journal.pone.0039336 (2012).

Reviewers' comments:

Reviewer #1 (Remarks to the Author):

The authors have adequately responded to the comments presented.

Minor issues:

The authors have performed RNAseq analysis of retinas, but description of the methods and information of the statistical testing (incl. Tables1-2) have not been included.

Figure S6e. Instead of an arrowhead, the figure has an arrow.

Reviewer #2 (Remarks to the Author):

The authors have address all my concerns to satisfaction. The rebuttal was comprehensive and well documented. Equally the new version of the manuscript is excellent. I no longer have concerns.

Reviewer #3 (Remarks to the Author):

The revised manuscript by Dudiki et al. aims to investigate microglial mechanosensing and it's role in regulating vascular patterning in the retina. The effects of Kindlin3 on microglia/macrophage cell spreading, TGF β 1 production, and subsequent vascular spreading are quite strong and interesting. However, the link with machanosensation is still weak. While the authors do provide substantially more data, many experiments do not sufficiently address reviewers' concerns. Remaining major concerns are outlined below:

- 1.) Unless more convincing experiments are performed, the authors should consider omitting data regarding tissue stiffness sensing. While appreciated that it is a difficult question to answer, the current experiments in Figs 1 and 2 do not effectively address this mechanism. In contrast, the experiments demonstrating microglia/macrophage contractility, which could be independent of

tissue stiffness sensing, can modulate TGF β 1 production are quite interesting and the potential link with vascular sprouting are better supported by the data.

2.) The authors make an additional attempt to link changes in microglial morphology with tissue stiffness *in vivo* using chondroitin sulfate (CS) in the revision. The authors state that CS makes the tissue softer, without directly altering microglial status. Unfortunately, CS has been shown to directly activate microglia through CD44 (see Rolls et al. PLOS Medicine 2008). This seems to be apparent from the experiments in this paper, too, where there are amoeboid microglia, indicative of reactive gliosis, in Fig 1h. Therefore, it is not possible to determine whether the effects are directly due to changes in tissue stiffness. Additionally, if CS does specifically affect microglial polarity via changes in tissue stiffness, it is unclear why changes in cell polarity are not uniform across the retina since CS delivery is throughout the retina. Finally, the representative images after CS treatment (Fig 1) do not represent the quantified data (Fig 1G). There appear to be larger morphological differences between the intermediate control and CS than the deep control and CS.

3.) The authors suggest a model that includes integrin signaling in regulating Kindlin3 (Fig 8); however, the authors provide evidence in the manuscript that Kindlin3 effects are independent of integrin signaling. This raises the question of how Kindlin3 is acting without binding integrin, if the model is through mechanosensation of tissue stiffness. It seems just as plausible that Kindlin 3 is acting through a different mechanism (the authors mention PIP2 in the discussion) to modulate cell contractility and TGF β 1 production, which was not explored in this manuscript.

4.) If microglial TGF β 1 is modulated by tissue stiffness to ultimately regulate SMAD signaling, microglial TGF β 1 expression and SMAD should differ across the different layers of the retina in a tissue stiffness dependent manner. This seems like a relatively straightforward experiment to perform, which could better link the mechanism with tissue stiffness.

5.) It remains unclear how the authors arrived at Kindlin3 as a candidate molecule if they did not initially see a phenotype in the integrin KO's given that this is the only known role of Kindlin 3 is "to bind and activate integrins". This requires more justification.

6.) The authors still cannot conclude that their effects on vasculature sprouting are microglia specific. Despite their rebuttal, I have to push back and inform the authors that there are existing, long-lived macrophages in the retina and brain vasculature called perivascular macrophages, which also express CX3CR1 (see Faraco et al. J Mol Med 2017 for a nice review of these cells) and are also sensitive to microglia ablation methods. If one is to go to higher magnification in the CX3CR1-EGFP reporter mice, these cells will be visualized. They express less CX3CR1 compared to microglia so they are more difficult to visualize at lower magnification. Given the vascular sprouting phenotype, it is important to address the possibility that this could be a perivascular macrophage-mediated effect or refine the conclusions to include this possibility.

7.) The authors provide Iba1 and TMEM119 staining to demonstrate microglia-specific expression of Kindlin3. However, Iba1 is broadly expressed by all macrophages and the TMEM119 staining is poor quality with very high background so it is difficult to assess specificity. Additionally, it is difficult to assess the Kindlin3 immunostaining and, at least at low magnification, it does not seem to 100% colocalize with microglia. The Venn diagrams are not a typical way in which one presents colocalization data and are uninformative. A more quantitative assessment of colocalization is necessary.

8.) The authors did not observe changes in angiogenesis-related molecules or in TGF β 1 signaling in the K3KI mice by RNAseq in Supplementary Figure 4. This is concerning and detracts from the rest of

the data. The authors also don't report any of the genes that were differentially expressed, which makes it difficult to interpret.

9.) N's are still largely defined by fields of view vs. separate experiments or animals. This still requires revision to add power to the statistical analyses.

10.) It is unclear how the authors measure cell polarity, beyond stating that is ratio of the length: width of the microglia briefly in the figure legend. What did the authors define as the width and length of a cell? What software was used to measure these parameters? Etc. This is not outlined in the methods. Given it is a major analysis performed in the paper, this is important to add to the methods section.

11.) It would be helpful if the authors stated what age mice they are using for all experiments. There are times where this information is omitted and could be useful for interpreting experimental results.

Response to the Reviewers and Editor

To address the reviewer's and editor's concerns we have performed several additional experiments and added new Figures 2d, g-i, S1b-e, S2a-f and S4c-e.

The only experiment that is not possible to perform is live imaging of microglia migrating through the layers of retina. The process takes over 7 days and we are not aware of any methods allowing such visualization without eye damage and microglial activation. The reviewers are aware of these limitations. However, since this process is highly conserved we perform day by day analysis of retinal microglia (supplementary Fig.1a,b) to show the process of migration from superficial to intermediate to deep layer with quantification of microglia at each stage. We also show microglial polarization and changes in TGFbeta (latent and active form) in the substrates of different stiffness in vitro. Please note that the conversion of high resolution images in our figures into PDF file format resulted in some loss of resolution and could occasionally be seen as grainy or pixelated images when zoomed. High quality images in TIFF or any other format preferred by the journal will be provided if required.

Reviewer 1:

1. The authors have performed RNAseq analysis of retinas, but description of the methods and information of the statistical testing (incl. Tables1-2) have not been included.

We have included the method and information on statistical testing used for the RNA transcriptome analysis into the methods section.

2. Figure S6e. Instead of an arrowhead, the figure has an arrow.

The arrow has been changed to an arrow head.

Reviewer 3:

1. Unless more convincing experiments are performed, the authors should consider omitting data regarding tissue stiffness sensing. While appreciated that it is a difficult question to answer, the current experiments in Figs 1 and 2 do not effectively address this mechanism. In contrast, the experiments demonstrating microglia/macrophage contractility, which could be independent of tissue stiffness sensing, can modulate TGFβ1 production are quite interesting and the potential link with vascular sprouting are better supported by the data.

The most conventional way to show mechanosensing, which is used by many groups in many high profile papers (i.e. recently published manuscript in Nat Cell Biology on a related topic by Jaumouille et al <https://www.nature.com/articles/s41556-019-0414-2>, see file attached and the section on mechanosensing is highlighted) is to change the stiffness of the substrate/object and

to demonstrate the response to such change. These experiments are done *in vitro* and *ex vivo* (as in attached paper) since these systems permit accurate and well controlled changes in stiffness. We provide such an evidence using primary microglia in two distinct types of hydrogels with predetermined stiffness (Fig.2 a-c,n-p; and Fig.S4 a,b). The experimental procedures used for this part have been published by other groups. Our *in vivo* evidence of mechanosensing stands upon the following:

1) During development, microglia populate retinal layers of varying stiffness, finally reaching the outer nuclear layer (ONL) around P12-16. In Fig.S1 we provide day by day imaging of microglia populating the retinal layers, see images below. Note a new bar graph in Fig.S1b showing the time course of microglia distribution (provided for the reviewer and the Editor's convenience within this document). Since ONL layer is 10 times stiffer than the regular CNS environment (Fig.1a,b,c below) and it triggers microglial polarization. This exact process is modelled *in vitro* using primary microglia and two types of hydrogels of varying stiffness mimicking retinal layers as discussed above. The lack of Kindlin3 selectively affects microglial response to stiffness, but not their migration or population of the retinal layers (Fig.3a,b below). Thus, the retina itself represents an excellent model of increasing stiffness. Since microglia reach different layers on a precise timeline (intermediate layer at P12, deep layer is completely populated at P16), the effect of stiffness might be monitored *in vivo* using retinas from pups of different ages (Fig.S1a,b, Fig.1c,d and Fig.3a,b below).

2) Exposure to the stiff substrate *in vitro* results in microglial polarization and downregulation of TGFbeta in wild-type (WT), but not in Kindlin3-deficient microglia (Fig.2n-p and Fig.S4a,b below).

3) We included a new set of the results showing that *in vivo* TGFbeta is secreted by amoeboid but not by polarized microglia (new Fig.S2a,b,c), which mirrors *in vitro* results from Fig.2n-p and Fig.S4a,b, indicating that polarization of microglia results in downregulation of TGFbeta, both, *in vitro* and *in vivo*. Moreover, we compared TGFbeta in microglia between intermediate and deep (stiff) layer *in vivo* and show that unlike intermediate ramified microglia, deep microglia polarized on stiff ONL no longer expresses TGFbeta *in vivo* (Fig.S2b and quantification of microglial TGFbeta in Fig.S2c). This is the second set of experiments aimed to "*link the mechanism with tissue stiffness*". See response #4 for additional details.

3) We included new results demonstrating that in our retinal model microglia is the main source of TGFbeta (which is in agreement with other studies in CNS). The knockout of TGFbeta in microglia diminished total TGFbeta signaling (as measured by phosphoSMAD3 levels) by 90% (Fig S2 and Fig 2d, g-i). Since TGFbeta is secreted in a latent form, which, in turn, is deposited

into the matrix and then activated, phosphorylation of downstream target SMAD3 demonstrates an activation of TGFbeta signaling within cells.

4) The downregulation of TGFbeta occurs *in vivo* based upon the spatiotemporal analysis of TGFbeta signaling within the retinal layers of varying stiffness (Fig.2e,f- the experiments also requested by the reviewer to “link the mechanism with tissue stiffness”). TGFbeta signaling is downregulated specifically within the deep layer (and not the intermediate layer), and only at P16, when microglia reach the stiff ONL (solid blue bar in Fig.2f). We now added a separate analysis of pSMAD3 in microglial cells (in addition to endothelium). It is well accepted that microglia expresses both, TGFbeta and its receptors, therefore, pSMAD3 levels in microglia reflect TGFbeta signaling in these cells. Our new data in Fig.S2d,e show that TGFbeta signaling in polarized microglia is 4 fold lower compared to nonpolarized cells (again, *in vivo* data mirror *in vitro* results). We also performed a comparative analysis of TGFbeta signaling between intermediate and deep microglia to show ~3 fold reduction in deep microglia vs intermediate one (new Fig.2d and Fig.S2f). This change is comparable to the reduction observed in microglia-specific TGFbeta KO (new Fig.2g-i).

Together, our *in vivo* and *in vitro* results show that microglial polarization and TGFbeta expression and signaling are controlled by tissue stiffness, i.e. mechanosensitive.

4) Using a recently-developed method of softening tissues in the CNS, we show that microglial polarization in WT mice is reversed by lowering the stiffness, whereas no response to stiffness changes was detected in Kindlin3-deficient microglia (the results were discussed in our previous submission).

We believe that in our manuscript, each conclusion is tightly linked to *in vivo* results and the point on mechanosensing is strongly supported by a combination of *in vitro* and *in vivo* experiments. We regret that some of our results were not fully appreciated by the reviewer, such as the data within Fig.2e and 2f showing spatiotemporal analysis of TGFbeta signaling *in vivo* within the layers of different stiffness, and Fig.3f,g showing comparison of WT microglia responses to chondroitin sulfate to that of Kindlin3-deficient microglia. Our results presented in Fig.1g were interpreted by the reviewer as a quantification of microglial response to chondroitin sulfate, whereas this panel actually shows the results of atomic force microscopy documenting the changes in retinal stiffness after chondroitin sulfate treatment. We did our best to clarify these issues below. We modified our results and discussion to make these points clear, especially the point about retinal layers as a model of varying stiffness. Additional controls and data are included in the revised manuscript as detailed below.

Microglia migrate through the stiffness gradient during retinal development.

Supplementary figure 1

Figure 1

Figure 3

K3 deficient microglia cannot sense and respond to matrix stiffness

Supplementary Figure 4a

Figure 2a.n

Regulation of microglial TGFβ1 by substrate stiffness

➤ K3 deficient microglia are unresponsive

Supplementary Figure 4a,b

Figure 3

Altering retinal stiffness ex vivo

2. The authors make an additional attempt to link changes in microglial morphology with tissue stiffness *in vivo* using chondroitin sulfate (CS) in the revision. The authors state that CS makes the tissue softer, without directly altering microglial status. Unfortunately, CS has been shown to directly activate microglia through CD44 (see Rolls et al. *PLOS Medicine* 2008). This seems to be apparent from the experiments in this paper, too, where there are amoeboid microglia, indicative of reactive gliosis, in Fig 1h. Therefore, it is not possible to determine whether the effects are directly due to changes in tissue stiffness.

The use of Chondroitin sulfate (CS) to modulate CNS stiffness is one of several approaches used in our manuscript. It was requested by the reviewer to alter the stiffness in the retina and to assess the changes within microglia. Based on careful analysis of the literature, CS treatment is the only current method available that is suitable for these experiments and validated for use in live neural tissues. The use of this method to evaluate *in vivo* stiffness in the CNS was demonstrated by Koser et al., *Nature Neuroscience*, 2016 (PMID: 27643431). However, in our study we use 6 and 4 hour incubation with chondroitin sulfate instead of 24 hours precisely to avoid microglia activation as shown below. As was pointed out by the reviewer, the timing and the consequences of possible microglial activation by CS *in vitro* were analyzed in Rolls et al., *PLOS Medicine*, 2008 (PMID: 18715114). This manuscript shows that microglia cultured *in vitro* on CS-coated coverslips for 48hrs undergo changes in morphology. They state “microglia cultured on CS exhibited a “fried egg” morphology, with flattened and thickened membrane processes and a larger cell body (mean \pm SD $37 \pm 10 \mu\text{m}$ in CS-cultured microglia versus $21 \pm 5.6 \mu\text{m}$ on a PDL base), a morphology associated with microglial activation”. At the same time, it was shown that CS did not increase expression of any microglial activation markers, i.e. NO, TNF, or BDNF, when microglia were plated on CS. Levels of these markers were upregulated only upon treatment with

lipopolysaccharide (LPS) as a positive control. As a mechanism for CS-induced activation, the manuscript proposes an IGF-1-dependent autocrine loop, and this conclusion is based upon mRNA analysis. However, no secreted IGF-1 was detected prior to 18hrs of CS treatment. In our experiment, we applied CS for the shortest possible time of 6 hours (similar changes in polarization were detected even after 4 hours) and monitored changes in retinal stiffness by atomic force microscopy as shown in Fig.1g. To confirm that this short-term application of CS does not substantially affect microglial morphology, we stained microglia with CD68, which is a lysosomal protein strongly expressed by activated microglia and macrophages, but only weakly expressed by resting microglia (see <https://www.abcam.com/cd68-antibody-ab125212.html> for references and examples). The settings of this experiment is similar to the one mentioned by the reviewer. CS treatment for 6 hours had no effect on CD68 expression (the level remained very low, thereby showing the lack of activation) in primary microglial cultures (see Supplemental Fig 1c,d). In this experiment to achieve an increase in CD68 we have used strong microglial agonist such as C5a as positive control/activator (Fig.S1c,d). As it can be seen from Fig.S1d, only C5a after 18 hours treatment promotes an increase in CD68, whereas CS at 6 hours time point has no effect. Moreover, in these settings we observed that both, nonpolarized as well as polarized cells are activated by C5a (Fig.S1d) and the loss of cell polarity was not a prerequisite for microglial activation.

We also repeated CS treatment of whole retinas and carefully monitored microglial activation by CD68 staining. As anticipated based on analysis of the literature, and as shown in Supplementary Figure 1e, 6 hours of CS treatment was sufficient to decrease retinal tissue stiffness (Fig.1g) and to induce microglial depolarization (Fig.1i, blue bars), but it was not sufficient to increase the expression of microglial activation marker CD68 (Fig.S1e). Thus, treatment with CS for 6 hours is not sufficient to promote microglial activation, which is a process requiring at least 18 hours of exposure as demonstrated by Rolls et al. (*PLOS Medicine*, 2008).

Moreover, in our manuscript the same treatment with CS was also applied to Kindlin3-deficient retinas. These data are shown in Fig.3f and quantification of polarized microglia is shown in comparison to WT retinas in Fig.3g. It should be noted that the activating component (if any) is similar for these two models. However, as seen in Fig.3g, while CS depolarizes WT microglia, it has no substantial effect on Kindlin3-deficient microglia.

The abovementioned references are now cited in the manuscript and the respective statements have been added.

Additionally, if CS does specifically affect microglial polarity via changes in tissue stiffness, it is unclear why changes in cell polarity are not uniform across the retina since CS delivery is throughout the retina. Finally, the representative images after CS treatment (Fig 1) do not represent the quantified data (Fig 1G). There appear to be larger morphological differences between the intermediate control and CS than the deep control and CS.

CS penetrates the retina layer by layer, thereby affecting the surface layers first and reaching the deep layer later, therefore the changes are not expected to be uniform. Nevertheless, the changes occur in both intermediate and deep layers, as is shown by the representative images in Fig.1h. It appears that the reviewer is mistaken since Figure 1g is not the quantification of the images from Fig.1h, it summarizes atomic force microscopy data.

The change in stiffness is in fact documented in our experimental system not merely stated in the text. Quantification of the effects of CS on cell polarity index (calculated as the ratio of longest to shortest axis) is shown in Fig. 1i. We use polarity index since this is a parameter that closely follows changes in mechanical stiffness, both *in vitro* and *in vivo*. The changes within the intermediate layer are indeed apparent since CS penetrates the retinal surface layers faster than

the deep layers. The change in polarity index for the intermediate layer is from ~1.7 to 1.5 (Fig.1i, orange bars as indicated), whereas for the deep layer the change in cell polarity is much more dramatic, from ~3.5 to 1.7 (Fig.1i, blue bars).

3. The authors suggest a model that includes integrin signaling in regulating Kindlin3 (Fig 8); however, the authors provide evidence in the manuscript that Kindlin3 effects are independent of integrin signaling. This raises the question of how Kindlin3 is acting without binding integrin, if the model is through mechanosensation of tissue stiffness. It seems just as plausible that Kindlin 3 is acting through a different mechanism (the authors mention PIP2 in the discussion) to modulate cell contractility and TGFB1 production, which was not explored in this manuscript.

Kindlin3 is an adaptor protein that directly binds and regulates integrins, which is its main function. Therefore, our model includes integrin. As stated in the manuscript and in cited papers and reviews, Kindlin has other binding partners. These include membrane components, which are recognized via a PH domain and a pseudo PH domain (see Meller, Rogozin et al., 2015, MBC) and cytoskeleton, including paxillin family members. The reviewer is correct that experiments with a specific knock-in mutant and individual integrin knockouts demonstrate that the effect on vasculature is integrin-independent. This highlights a significant message, since all of the other functions of Kindlin are believed to be primarily integrin-dependent.

We demonstrated that Kindlin-mediated mechanosensing is also dependent upon cell contractility (Fig.7a), that it can be rescued by loosening myosin cytoskeleton with blebbistatin (Fig.7b and Fig.S10d-f), and that it is also dependent upon ERK activation (Fig.7d-f). The entire chain of events leading to normalization of TGFbeta expression was rescued by interference with cytoskeletal connections and ERK over-activation, which implicates a Kindlin3-cytoskeletal link. At the same time, there is no doubt that binding to PIP2 through the Kindlin3 PH domain is crucial for Kindlin3 functions, since PH domain deletion phenocopies a complete Kindlin3 knockout as we have previously shown (Meller et al., 2012). Kindlins lacking membrane-binding domains are not properly localized within the cell, and in many cases are poorly expressed. We have modified our discussion to make these issues clear.

4. If microglial TGFβ1 is modulated by tissue stiffness to ultimately regulate SMAD signaling, microglial TGFβ1 expression and SMAD should differ across the different layers of the retina in a tissue stiffness dependent manner. This seems like a relatively straightforward experiment to perform, which could better link the mechanism with tissue stiffness.

This is a very good point since the retina itself is composed of layers of varying stiffness (Fig.1) and migrating microglia (Figs.1,3 and S1a,b) experience these changes in mechanical stiffness. Please see our response to #1 above.

We included new results showing that in vivo TGFbeta is secreted by amoeboid but not by polarized microglia (new Fig.S2a,b,c), which mirrors in vitro results from Fig.2n-p and Fig.S4a,b,

indicating that polarization of microglia results in downregulation of TGFbeta, both, in vitro and in vivo. We included new comparison of microglial TGFbeta between intermediate and deep (stiff) layer in vivo to show that unlike intermediate ramified microglia, deep microglia polarized on stiff ONL no longer expresses TGFbeta in (Fig.S2b and quantification of microglial TGFbeta in Fig.S2c). This is the second set of experiments aimed to “*link the mechanism with tissue stiffness*”.

Supplementary Figure 2

Since microglia is the major source of TGFbeta1 that is secreted in a latent form, which, in turn, is deposited into the matrix and then activated, pSMAD3 demonstrates an activation of TGFbeta signaling. We measured in vivo levels of phosphoSMAD3 (the readout of TGFbeta signaling as pointed by the reviewer) within the each layer. This microglial regulation of TGFβ1 expression in response to tissue stiffness i.e microglial bipolarization on stiff ONL at P16 results in a 3-fold reduction in endothelial pSMAD3 within the deep vascular layer. At the same time, no changes occurred in the intermediate layer of lower stiffness, where microglia remained ramified/unpolarized (Fig.2e,f). This reduction in TGFβ1 signaling in deep vasculature coincided with a termination of its further growth into the ONL (Fig.1), thereby implicating microglial TGFβ1 in the restriction of excessive retinal vasculature.

Figure 2e

Microglia specific knockout of TGFB1 (in CX3CR1-cre;TGFB1^{ff} mice) results in ~90% loss of this phosphoSMAD3 localization to endothelial cells (Fig.2g-i) confirming microglial TGFB1 dependent signalling. Further, the same is seen in microglia specific K3 and TGFB1 double knockout mice (CX3CR1-cre;K3^{ff}/TGFB1^{ff} mice). Tamoxifen treatment of these mice results in >80% loss in phosphoSMAD3 levels, thereby rescuing the vascular phenotype of K3KO mice (Fig.6d-f).

Figure 2

Figure 6

Since TGFβ1 can also act through autocrine signaling, comparison of amoeboid/unpolarized and polarized microglia within the retina showed >75% decrease in phosphoSMAD3 localization to the nuclei of polarized microglia (Fig. S2d-f below and Fig. 2d). In sum, we provide *in vivo* data to conclusively link our mechanism with tissue stiffness.

Supplementary Figure 2

5. It remains unclear how the authors arrived at Kindlin3 as a candidate molecule if they did not initially see a phenotype in the integrin KOs given that this the only known role of Kindlin 3 is “to bind and activate integrins”. This requires more justification.

The main justification for our study is the presence of a dramatic vascular and microglial phenotype in Kindlin 3-deficient retinas. Our initial analysis was focused on Kindlin3 and not on integrins, which are widely expressed and exhibit substantial functional redundancy. At the same time, the lack of Kindlin3 leads to a devastating human disorder (lethal in most cases) characterized by bleeding as well as immune and bone problems. Deletion of Kindlin-1 paralog (expressed in skin) also causes a human disorder associated with premature aging, known as Kindler syndrome. Thus Kindlins are clearly important for human physiology/pathology. The main justification to explore the mechanism of Kindlin3 function in microglia is based upon severe cerebrovascular complications of unknown nature observed in our Kindlin3-deficient patients (Malinin et al., *Nature Medicine*, 2009; PMID: 19234460). Based upon these previous studies, the standard care for Kindlin-3-deficient patients is irradiation and complete bone marrow replacement at an early age, which is a procedure that affects microglia. Therefore, our studies on Kindlin-3 in microglia were performed in mouse models and are presented in the current manuscript. We have added respective statements to the introduction.

6. The authors still cannot conclude that their effects on vasculature sprouting are microglia specific. Despite their rebuttal, I have to push back and inform the authors that there are existing, long-lived macrophages in the retina and brain vasculature called perivascular macrophages, which also express CX3CR1 (see Faraco et al. J Mol Med 2017 for a nice review of these cells) and are also sensitive to microglia ablation methods. If one is to go to higher magnification in the CX3CR1-EGFP reporter mice, these cells will be visualized. They express less CX3CR1 compared to microglia so they are more difficult to visualize at lower magnification. Given the vascular sprouting phenotype, it is important to address the possibility that this could be a perivascular macrophage-mediated effect or refine the conclusions to include this possibility.

We appreciate this comment especially since these perivascular macrophages (PVMs) cells are also of myeloid origin expressing K3 and function similar to microglia, they could possibly contribute to the phenotype in our models. However, as shown in the figure below, at the age of P21 that is relevant to this study there were negligible numbers (<40 cells/retina) of PVMs identified by their specific marker MHCII. Importantly, this small population of PVMs were mostly observed at the optic nerve or periphery of the retina. PVMs were usually not observed within the radius of 500-1500um from the optic nerve of the retina where neo-vascular sprouts and vascular sprouts exist in kindlin3 deficient mice. Hence, given the negligible numbers of these cells in normal retina at P21 in addition to their absence at sites of neo-vascular sprouts and vascular

lesions, their contribution to the robust abnormal vascular phenotype is unlikely. We have now included this in our discussion.

Our data (below) support the results of several other groups that characterized and quantified PVMs in normal retinas. These studies concluded that the PVM numbers are negligible compared to microglia, especially at the age we use in our study (the peak of our phenotype occurs at P16-P21). Below are the references to studies in retinas:

1) Xu et al. (*Invest Ophthalmol Sci* 2007) show **<15 PVMs per whole retina** in 2 weeks old mice. At 4 weeks there were <60 cells/retina.

2) Similarly, Leehman et al. (*Neurobiol Dis* 2010) show 95 ± 42 cells per quiescent retina in adult mice (N = 14).

3) Another study by Koren et al. (*Sci Rep* 2017) also shows PVMs constitute a very small fraction of microglia in normal adult retinas.

Below, we show new staining of entire retina from optic nerve to periphery using a specific marker for peripheral macrophages, MHCII (red) in parallel with microglia (white). Note the negligible amounts of perivascular macrophages restricted to the optic nerve area, which is in agreement with the studies cited above.

7. The authors provide Iba1 and TMEM119 staining to demonstrate microglia-specific expression of Kindlin3. However, Iba1 is broadly expressed by all macrophages and the TMEM119 staining is poor quality with very high background so it is difficult to assess specificity. Additionally, it is difficult to assess the Kindlin3 immunostaining and, at least at low magnification, it does not seem to 100% colocalize with microglia. The Venn diagrams are not a typical way in which one presents colocalization data and are uninformative. A more quantitative assessment of colocalization is necessary.

We have included new higher resolution figures of TMEM119 staining showing co-localization with CX3CR1 GFP (Fig.S4d).

Higher magnification images of Kindlin3 staining showing co-localization with CX3CR1 GFP have been included as well (Fig.S4c). Similar results have been previously shown in Meller et al., *JCI Ins*, 2017.

Venn diagrams to show co-localization are used by many groups in highly-cited journals (e.g. *Neuron*; 85(4): 833–846; PMID: 25640077 – specifically for microglia markers). These diagrams clearly convey the necessary message, especially with the percentages shown on the diagram. However, upon the critique of the reviewer, we have replaced some Venn diagrams with a table summarizing the same numbers (Fig.S4e).

Supplementary Figure 4

8. The authors did not observe changes in angiogenesis-related molecules or in TGFβ1 signaling in the K3KI mice by RNAseq in Supplementary Figure 4. This is concerning and detracts from the rest of the data. The authors also don't report any of the genes that were differentially expressed, which makes it difficult to interpret.

The point of whole-retina RNA transcriptome analysis is to show global changes in the visual system upon Kindlin3 deletion in microglia. As stated in the manuscript and in cited publications, microglia represent a very minor cell type in the context of the entire retina. This it is anticipated

that reduction in microglial TGFbeta, which occurs only within the stiffest layer of the retina, cannot be detected at the level of entire retina. This result was presented as a rationale to perform the search using isolated microglia and to probe individual candidate genes.

9. N's are still largely defined by fields of view vs. separate experiments or animals. This still requires revision to add power to the statistical analyses.

As suggested, we have revised the graphs and their statistics to show the values of individual animals.

10. It is unclear how the authors measure cell polarity, beyond stating that is ratio of the length: width of the microglia briefly in the figure legend. What did the authors define as the width and length of a cell? What software was used to measure these parameters? Etc. This is not outlined in the methods. Given it is a major analysis performed in the paper, this is important to add to the methods section.

We expanded our Methods section to include the requested information. Cell polarity was measured using Image J. The following steps were involved: (i) Image processing to separate microglial cells from background; (ii) Construction of a skeleton to represent the spatial structure of cell bodies and branches if required; (iii) Generation of dendritic tree area to identify the longest and widest axis; (iv) Measurement of microglial length in pixels, including cell body and the branches that are along the longest dendritic tree axis. Similarly, the width was measured along the widest axis; (v) Division of length by width to get the ratio, i.e. polarity.

11. It would be helpful if the authors stated what age mice they are using for all experiments. There are times where this information is omitted and could be useful for interpreting experimental results.

We now show the age of mice on the images themselves for the convenience of the reviewers and readers. This information is also presented in figure legends/results.

REVIEWERS' COMMENTS:

Reviewer #3 (Remarks to the Author):

In the revised manuscript, it is clear that the authors' put a lot of work into addressing remaining concerns. The new data showing TGF β 1 and SMAD expression in different layers of the retina are particularly nice and the new experiments to address a potential contribution of PVMs are appreciated. The major critique of this paper remains that the authors do not provide definitive in vivo evidence that microglial sensing of tissue stiffness directly modulates their morphology and TGF β 1 production nor do they provide a direct role for microglial kindlin 3 in sensing tissue stiffness in vivo. The data supporting this mechanism, while quite interesting, remain correlative so the conclusions should be toned down, including the title. It detracts from their strongest data showing a role for kindlin 3 in regulating cell contractility and the vasculature. This and other remaining concerns are outlined below:

1. The authors provide correlative evidence in vivo that as stiffness changes through the tissue, microglia morphology also changes. They also provide nice new data showing a correlation between tissue stiffness and TGF β signaling. The authors provide more direct in vitro evidence for a role for microglial kindlin 3 in sensing changes in extracellular stiffness, but microglia are quite different cells in vitro compared to in vivo. The last half of the paper regarding the role of microglial kindlin 3 in regulating cell contractility are quite convincing and nice. Therefore, this major concern can be adequately handled by revising the text to emphasize the contractility data towards the end of the manuscript and toning down the language regarding the mechanosensing mechanism. While intriguing, the in vivo data supporting this mechanism are correlative.
2. There are still concerns regarding the chondroitin sulfate (CS) experiments. Apologies for mis-referencing a figure in the last review, which caused some confusion (referenced 1g, intending to reference 1i). The representative images after CS treatment (Fig 1h) still do not appear to represent the quantified data (Fig 1i). There appear to be larger morphological differences between the intermediate control and CS than the deep control and CS in the images in panel h, which is not reflected in the quantification in panel i. Moreover, the microglia in the CS treated retinas appear to be amoeboid in the intermediate layer, indicative of reactive gliosis, in Fig 1h. While the authors suggest that CS does not activate microglia by performing CD68 immunofluorescence, this is not sufficient alone to conclude microglia are not activated. Moreover, the CD68 immunofluorescence shown in supplemental figure is from a P21 retina in the deep layer. This is inconsistent with data from figure 1i, where microglia cell polarity was quantified from P16 retinas in both layers.
3. As stated in the author's rebuttal, the main function of Kindlin3 is to bind to and regulate integrins. However, the author's provide evidence that the role of Kindlin3 in their model is integrin independent (Fig 2). Therefore, it is unclear why integrins are still included in their model in Fig. 8?

Point-by-point response to issues raised by our referees (the referees' comments are provided in italics).

Dear Editor,

Below are point-by point edits to address final requests from the Editor and the Rev.3

From Editor: “In particular, we would require appropriate toning down of conclusions regarding the potential correlative nature of current results microglia mechanoresponsiveness, acknowledging caveats to the CS perturbations, and potential roles of kindlin3 in mechanosensation”.

Rev.3 1. *“The authors provide correlative evidence in vivo that as stiffness changes through the tissue, microglia morphology also changes. They also provide nice new data showing a correlation between tissue stiffness and TGF β signaling. The authors provide more direct in vitro evidence for a role for microglial kindlin 3 in sensing changes in extracellular stiffness, but microglia are quite different cells in vitro compared to in vivo. The last half of the paper regarding the role of microglial kindlin 3 in regulating cell contractility are quite convincing and nice. Therefore, this major concern can be adequately handled by revising the text to emphasize the contractility data towards the end of the manuscript and toning down the language regarding the mechanosensing mechanism. While intriguing, the in vivo data supporting this mechanism are correlative.”*

As per suggestions from the rev. 3 we toned down our conclusions regarding mechanosensing in vivo. We now directly specify what was done in vivo and what was substantiated *in vitro* such as experiments in two types of hydrogels of various stiffness etc. This will provide the readers with an opportunity to judge for themselves and make their own conclusions from our results. Our abstract and Title were modified as well. The revisions are in blue.

Specifically, we modified the results section and discussions of the data including experiments with chondroitin sulfate tissue softening in vivo. We modified the interpretation of these results and discussed the possibility of microglial activation by chondroitin sulfate as requested. We are in agreement with the reviewer that microglia needs to be studied in vivo due to its activation and, therefore, (as you can see) our manuscript takes an advantage of multiple in vivo models, more so than the majority of the published manuscripts on microglial function. Treatment with CS remains to be the one and only available approach of altering tissue stiffness in vivo within CNS and seems to be of importance for the scientific community.

Also as suggested, we emphasized our data regarding microglia contractility in the absence of Kindlin3 as an important feature affecting our phenotype. For this part of the mechanism (as you also well know) we have a series of rescue experiments (Figure 7) allowing us to directly link Kindlin3, microglial bipolarization and its direct effect on TGFbeta1 expression as a function of substrate stiffness (Fig.2 and 7).

2. *The representative images after CS treatment (Fig 1h) still do not appear to represent the quantified data (Fig 1i). There appear to be larger morphological differences between the intermediate control and CS than the deep control and CS in the images in panel h, which is not reflected in the quantification in panel i. Moreover, the microglia in the CS treated retinas*

appear to be amoeboid in the intermediate layer, indicative of reactive gliosis, in Fig 1h. While the authors suggest that CS does not activate microglia by performing CD68 immunofluorescence, this is not sufficient alone to conclude microglia are not activated. Moreover, the CD68 immunofluorescence shown in supplemental figure is from a P21 retina in the deep layer. This is inconsistent with data from figure 1i, where microglia cell polarity was quantified from P16 retinas in both layers.

As requested images in Fig 1h have been replaced with a more representative image. We also added a new set of microglial morphology quantifications in retinas upon treatment with chondroitin sulfate to reflect “larger morphological differences” noted by the Rev.3. The images are shown in Fig.1h, quantification of stiffness in Fig.1g and quantification of cell polarity in Fig1i. New quantification of the length of microglia processes is in SFig.1f and numbers of processes are shown in SFig.1g. Note that similar parameters were requested by this reviewer in respect to other experiments to fully illustrate microglial shape change in vivo. Please, note that as described in Figure legends we show flattened images of microglia and thinner processes are obvious at higher magnification. The text has been modified accordingly to note that microglial activation might still occur despite low level of CD68 activation marker.

3. As stated in the author’s rebuttal, the main function of Kindlin3 is to bind to and regulate integrins. However, the author’s provide evidence that the role of Kindlin3 in their model is integrin independent (Fig 2). Therefore, it is unclear why integrins are still included in their model in Fig. 8?

We modified our cartoon based on the rev.3 comments. We mark an integrin binding site on Kindlin3 to illustrate the nature of our main Kindlin 3 knockin mouse model and indicate that “integrin binding not involved”. The image of integrin was moved to the background and now appears as a part of the cell membrane. If this is not satisfactory to the Editors, we are OK with deleting the image of integrin completely.

We do hope that the Editorial Board will find these modifications sufficient. Our abstract and title have been modified and we will gladly solicit an advice from the Editorial Board to further increase the clarity and impact of our work as reflected in our abstract and title.